# Self-Interpretable Model with Transformation Equivariant Interpretation

**Yipei Wang, Xiaoqian Wang** *
Elmore School of Electrical and Computer Engineering
Purdue University
West Lafayette, IN 47907
`wang4865@purdue.edu, joywang@purdue.edu`

## Abstract

With the proliferation of machine learning applications in the real world, the demand for explaining machine learning predictions continues to grow especially in high-stakes fields. Recent studies have found that interpretation methods can be sensitive and unreliable, where the interpretations can be disturbed by perturbations or transformations of input data. To address this issue, we propose to learn robust interpretations through transformation equivariant regularization in a self-interpretable model. The resulting model is capable of capturing valid interpretations that are equivariant to geometric transformations. Moreover, since our model is self-interpretable, it enables faithful interpretations that reflect the true predictive mechanism. Unlike existing self-interpretable models, which usually sacrifice expressive power for the sake of interpretation quality, our model preserves the high expressive capability comparable to the state-of-the-art deep learning models in complex tasks, while providing visualizable and faithful high-quality interpretation. We compare with various related methods and validate the interpretation quality and consistency of our model.

## 1 Introduction

Deep learning (DL) models have been a great success in various domains of applications, including object detection, image classification, etc. However, many applications suffer from the overfitting problem, which is usually due to the lack of various training data. For scenarios with limited data access, data augmentation is usually applied to alleviate the overfitting problem. As one of the most simple but effective data augmentation methods, geometric transformation plays an important role in exploring the intrinsic visual structures of image data [44, 34]. Transformation equivariance refers to the property that data representations learned from the model capture the intrinsic coordinates of the entities [15], i.e., transformations on the data will result in the same transformations to the model representations. Building transformation-equivariant DL models is desired in many kinds of applications, such as medical image analysis [11], reinforcement learning [27], etc.

Although DL models can exert excellent performance in various tasks, DL models are usually expressed as black boxes. Therefore, DL models can have great performance in complex tasks but lack an explanation of the results [12]. In low-risk tasks such as adaptive email filtering, the direct deployment of black-box models without reasoning might be acceptable. However, for high-risk decision-making tasks such as disease diagnosis and autonomous vehicles [18], the applied model needs to be more convincing than a black box. On the one hand, by faithfully explaining the model behavior, it can ensure the end user intuitively understands and trusts the DL model. On the other hand, the explanation of black-box models can provide insights into the relationship between input

---

*Corresponding author.

35th Conference on Neural Information Processing Systems (NeurIPS 2021).

and output, thereby improving model design. However, with the rapid growth in computational power, DL models are designed to be more and more complex to meet the performance [35], and the most advanced DL models can have billions of trainable parameters [36]. High complexity leads to the complete black box for human beings, which results in a lack of trust in the model. The demand for building more reliable and easy-to-understand DL models is growing rapidly.

Depending on the stages where predictions and interpretations are conducted, the methods can be divided into two opposing categories: self-interpretable models and post-hoc models [30]. Unlike post-hoc models, which generate interpretations to pre-trained black-box models, self-interpretable models aim to build models that are intrinsically interpretable themselves. The main difference between these two categories is that for post-hoc models, the interpretation and prediction are obtained in two different stages. The interpretation is separately obtained after the black-box models are trained. Therefore, interpretations obtained from post-hoc models are considered to be more fragile, sensitive, and less faithful to the predictive mechanism [1, 19, 50]. In contrast, self-interpretable models make interpretations at the same time as predictions, thus revealing the intrinsic mechanism of the models, and are thereby preferred by users in high-stakes tasks [40]. Besides, considering how powerful and common transformation can be in data augmentations, it is reasonable to take the robustness of interpretation to transformations into consideration when designing and evaluating the interpretations. Robust interpretation towards transformation implies two requirements: 1) the predictive mechanism indicated by the interpretation should remain the same after transformation (e.g., the highlighted region should remain the same despite the transformation); 2) the location of interpretation should change according to the transformation. These two requirements naturally lead to *transformation equivariance on interpretation*. The transformation-equivariance property will enhance robust and faithful interpretation, where the interpretation is aware of the transformations and preserves the predictive mechanism. This correspondence between transformation equivariance and faithfulness suggests that self-interpretable models may perform better than post-hoc models in transformation awareness given their higher faithfulness. And the experiments also demonstrate this.

Although self-interpretable models surpass post-hoc models in faithfulness and stability, there are non-negligible challenges in building self-interpretable models. First, the interpretations may need additional regularization to be in forms that are rational to humans. This process usually involves prior domain knowledge provided by human experts [21, 39]. Besides, since the interpretability is intrinsic, specific constraints are required in the models to ensure the interpretability. The prediction power of such models will be damaged since it is essentially adding constraints to optimization problems. It is acknowledged that the increase of the interpretation quality is likely to decrease the performance of prediction results [13, 31]. As a consequence, self-interpretable models are usually less expressive compared with black-box models, which can be interpreted by post-hoc models.

**Our Model:** In this paper, we develop a transformation-equivariant self-interpretable model for classification tasks. As a self-interpretable model, our method makes predictions and generates interpretations of the predictions at the same stage. In other words, the interpretations are directly involved in the feed-forward prediction process, and are therefore faithful to the final results. We name our method as SITE (Self-Interpretable model with Transformation Equivariant Interpretation). In SITE, we generate data-dependent prototypes for each class and formulate the prediction as the inner product between each prototype and the extracted features. The interpretations can be easily visualized by upsampling from the prototype space to the input data space.

Besides, we introduce transformation regularization and reconstruction regularization to the prototypes. The reconstruction regularizer regularizes the interpretations to be meaningful and comprehensible for humans, while the transformation regularizer constrains the interpretations to be transformation equivariant. We validate that SITE presents understandable and faithful interpretations without requiring additional domain knowledge, and preserves high expressive power in prediction.

We summarize the main contributions through this work as:

- To our best knowledge, we are the first to learn transformation equivariant interpretations.
- We build a self-interpretable model SITE with high-quality faithful and robust interpretation.
- SITE preserves the high expressive power with comparable or better accuracy than related black-box models.
- We propose *self-consistency score*, a new quantitative metric for interpretation methods. It quantifies the robustness of interpretation by measuring the consistency of interpretations to geometric transformations.

## 2 Related Work

Machine learning interpretation can have different goals, such as attribution [24], interpretable clustering [28], interpretable reinforcement learning [29], disentanglement [43], etc. Our method lies in the attribution category, thus we mainly review the related interpretation methods for attribution.

Attribution methods target at identifying the contribution of different elements in the prediction. Based on if the prediction and interpretation are obtained in the same stage, the methods can be divided into post-hoc interpretation and self-interpretable methods.

For post-hoc interpretation, the prediction results are obtained by a black-box model while the interpretation is obtained separately to explain the predictive mechanism of the black box. Among the different post-hoc interpretation techniques, backpropagation methods [60, 41, 52, 45, 37, 4], trace from the output back to the input to determine how the different elements in the input contribute to the prediction result. Class Activation Mapping (CAM) [60] visualizes the feature importance in convolutional neural networks by mapping the weights in the last fully connected layer to the input layer via upsampling. Score-CAM learns weighting scores for the activation maps by integrating the increase in confidence for an improved CAM visualization [52]. While for approximation methods [38], the interpretation is obtained by fitting an interpretable model to the black-box prediction around the target sample. Deconvolution methods [58] interpret a convolutional neural network via image deconvolution. For perturbation-based interpretation [32, 14], the methods interpret the feature importance by imposing perturbation to certain feature and checking the changes in the output. Moreover, Shapley values [24] have been used to calculate the feature importance due to the nice properties preserved by Shapley values. For post-hoc interpretation, since the prediction and interpretation are separated, the prediction can be obtained by a highly expressive black-box model to handle complex tasks. However, the post-hoc interpretation may not capture the true predictive mechanism of the black box and is less reliable [1, 19].

Different from post-hoc interpretation, self-interpretable models target at building white boxes that are intrinsically interpretable, which are able to conduct prediction and interpretation at the same time. A self-interpretable model preserves faithful interpretation since the model itself is a white box. However, the self-interpretation constraints can limit the expressive power, thus sacrificing prediction performance. For example, in order to build an interpretable decision set [22], there is a constraint on the number of rules for the sake of interpretation, which restricts its application to complex tasks. Recent models propose to build self-interpretable models with neural network [2, 3, 6, 17, 20, 53, 54] and kernel methods [7]. FRESH [17] focuses on the interpretability for natural language processing tasks. SENN [3], Concept Bottleneck Models [20] generate interpretations in high-level spaces instead of the raw pixel space. ProtoPNet [6] provides interpretations in the pixel space, but it focuses more on the local patches corresponding to the local areas of the image instead of the global interpretation. NAM [2] provides the same kind of interpretations as SITE. It combines neural networks with additive models to facilitate the self-interpretation via component function. But it decouples all pixels, which results in low expressiveness. Moreover, attention models have been widely used to build interpretable predictions [26]. However, recent works find that the interpretation via attention weights can fail to identify the important representations [42].

Different from the related works, our goal is to build a self-interpretable model that learns faithful interpretation and has high expressive power. Previously the transformation equivariance property has been studied in prediction via deep neural networks. Many recent studies integrate the transformation equivariance in object detection, with the goal of building convolutional neural networks that are equivariant to image translations. The models learn features equivariant to translation and rotation [56, 55, 10], 3D symmetries [51], and build sets with symmetric elements for general equivariant tasks [25]. Despite transformation equivariance in the prediction, these methods may not guarantee the transformation equivariance in the interpretation, i.e., the prediction mechanisms of transformed and untransformed inputs may be inconsistent. We thus introduce the interpretation equivariance to complement the prediction equivariance. To the best of our knowledge, we are the first to learn transformation-equivariant interpretations to ensure faithful and robust interpretation.

## 3 Building Transformation Equivariant Interpretation in SITE

In this section, we introduce the structure of SITE. For notations, all normal lowercase letters stand for numbers; all bold lowercase letters stand for tensors; all normal uppercase letters stand

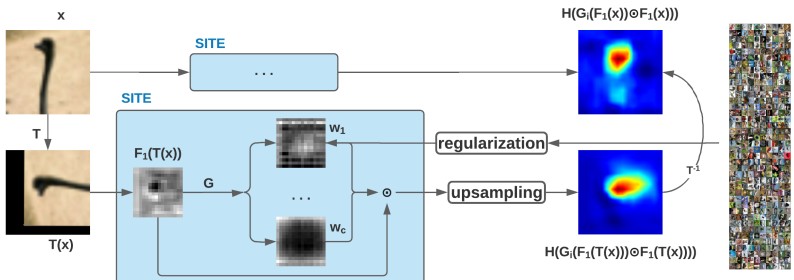

Figure 1: An illustration of our SITE model. SITE can take both original image $\mathbf{x}$ and transformed image $T(\mathbf{x})$ as input. The input is first fed to the feature extractor $F_1$, then SITE generates $c$ prototypes $\mathbf{w}_1, \cdots, \mathbf{w}_c$ through generator $G$. Finally, both the prediction and interpretation come from the Hadamard product between the latent representation $F_1(T(\mathbf{x}))$ and each prototype. The interpretation is obtained by upsampling the Hadamard product, and the prediction is obtained by the element-wise summation of it. SITE ensures transformation equivariant interpretation by constraining on the interpretations before and after transformation.

for operations (including functions, networks, etc); and all curly uppercase letters denote sets and families. Additionally, all Greek letters will be explained when they are introduced in context.

## 3.1 Formulation

For image classification tasks, suppose that $\mathbf{x} \in \mathbb{R}^p$ denotes the input image, one-hot vector $\mathbf{y} \in \{0, 1\}^c$ denotes the label, and $\hat{\mathbf{y}} \in [0, 1]^c$ denotes the predicted class probabilities. We clarify that $p$ is the product of the number of channels, the width, and the height of the image $\mathbf{x}$, while $c$ denotes the number of classes. Generally, a traditional classifier $F : \mathbf{x} \mapsto \hat{\mathbf{y}}$ can be decomposed into $F = F_2 \circ F_1$ with a feature extractor $F_1$ and a simple classifier $F_2$, where $F_1 : \mathbf{x} \mapsto \mathbf{z}$ and $F_2 : \mathbf{z} \mapsto \hat{\mathbf{y}}$. Here $\mathbf{z} \in \mathbb{R}^d$ denotes the extracted latent representations of $\mathbf{x}$, and usually has a lower dimension ($d < p$). The extractor $F_1$ usually consists of convolutional neural networks or ResNet structures, and $F_2$ consists of fully connected layers. The goal is to minimize the classification loss

$$\min_{F = F_2 \circ F_1} \mathbb{E}_{\mathbf{x} \in \mathcal{X}, \mathbf{y} \in \mathcal{Y}} L_{ce}(F(\mathbf{x}), \mathbf{y}), \tag{1}$$

where $\mathcal{X}, \mathcal{Y}$ are the input data set and the target set, and $L_{ce}$ denotes the cross-entropy loss function.

Traditional methods in (1) is not intrinsically interpretable w.r.t. the contribution of features in $\mathbf{x}$ to the prediction $\hat{\mathbf{y}}$. In order to address this, in SITE we build a generative model $G = [G_1, \cdots, G_c]$ that maps the latent representation $\mathbf{z}$ to $c$ *prototypes* $\{\mathbf{w}_i\}_{i=1}^c \subset \mathbb{R}^d$, where $\mathbf{w}_i = G_i(\mathbf{z})$. Each prototype corresponds to a specific class. We formulate the final prediction as the inner product of the latent representation $\mathbf{z}$ and each prototype $\{G_i(\mathbf{z})\}_{i=1}^c$. That is,

$$\hat{\mathbf{y}} = \sigma(G(\mathbf{z})^\top \mathbf{z}) = \sigma\left(\left[G_1(\mathbf{z})^\top \mathbf{z}, G_2(\mathbf{z})^\top \mathbf{z}, \ldots, G_c(\mathbf{z})^\top \mathbf{z}\right]^\top\right), \tag{2}$$

where $\sigma$ is softmax activation. The prediction $\hat{\mathbf{y}}$ is the similarity between the latent representation $\mathbf{z}$ and the generated prototype $G_i(\mathbf{z})$. Thus we have the modified classification loss

$$L_{cls} = \mathbb{E}_{\mathbf{x} \in \mathcal{X}, \mathbf{y} \in \mathcal{Y}} L_{ce}\left(\sigma\left(G(F_1(\mathbf{x}))^\top F_1(\mathbf{x})\right), \mathbf{y}\right). \tag{3}$$

Note that we formulate the prediction result for class $i$ as $\hat{y}_i = \sigma(G_i(\mathbf{z})^\top \mathbf{z}) = \sigma(\mathbf{w}_i^\top \mathbf{z})$. According to our formulation of $\hat{\mathbf{y}}$ in (2), we can explicitly capture the contribution of elements in $\mathbf{z}$ to the final prediction by the Hadamard product between $\mathbf{w}_i$ and $\mathbf{z}$ as $\hat{\mathbf{w}}_i = \mathbf{w}_i \odot \mathbf{z}$. Naturally we take $\hat{\mathbf{w}}_i$ as the *interpretation* of the $i$-th prediction result, such that the contribution of different elements to each prediction result is clear from the interpretation. For instance, $\hat{w}_i^j, i = 1, \cdots, c, j = 1 \cdots, d$ denotes the contribution of the $j$-th element of $\mathbf{z}$ to the $i$-th class. Based on our formulation of $\hat{\mathbf{w}}$, the interpretation from our model preserves the **completeness** property. That is, the summation of the importance scores of all features equals the prediction result. This is introduced as Proposition. 1 in [48], and also known as the **local accuracy** in [24]. This property assures that the interpretation is related to the corresponding prediction in the numerical sense.

The interpretation obtained from optimizing $L_{cls}$ ensures the faithfulness (i.e., which shows the true predictive mechanism of the model), but may not ensure that the interpretation is human-

understandable. In order to build high-quality interpretation, we propose to regularize the prototypes $G_i(\mathbf{x}), i = 1, \cdots, c$ with the following. For an input image $\mathbf{x}$, we enforce each generated prototype $G_i(F_1(\mathbf{x}))$ to be similar to its corresponding class's latent representation $F_1(\mathbf{x}_i)$:

$$L_1 = \sum_{i=1}^{c} \mathbb{E}_{\mathbf{x} \in \mathcal{X}, \mathbf{x}_i \in \mathcal{X}_i} L_{bce}\left(G_i(F_1(\mathbf{x})), F_1(\mathbf{x}_i)\right), \tag{4}$$

where $L_{bce}$ denotes the binary cross-entropy loss, and $\mathcal{X}_i \subset \mathcal{X}$ denotes the set of input data that belongs to the $i$-th class.

In addition, we propose to regularize on the *transformation equivariance* property of interpretation from our SITE model. Let $T_\beta$ denote pre-defined parametric transformations as described in [16]. We want SITE to learn interpretations that are equivariant to the transformations. Here $\beta \sim \mathcal{B}$ denotes the randomly sampled parameters from a pre-defined parameter distribution $\mathcal{B}$. This is because an affine transformation operator $T$ can be parameterized by an $3 \times 3$ matrix $\beta$. During the training process, we suppose that the random transformation $T_\beta$ is known and we can have access to its inverse $T_\beta^{-1}$. In the feed-forward process of training, we first transform the input image $\mathbf{x}$ by randomly sampled transformations $T_\beta(\mathbf{x}), \beta \sim \mathcal{B}$, then feed it to the model $G \circ F_1$. So the prediction result on the transformed image is $G(F_1(T_\beta(\mathbf{x})))^\top F_1(T_\beta(\mathbf{x}))$. The prototypes of the transformed input image $G(F_1(T_\beta(\mathbf{x})))$ can be transformed back by the inverse transformation $T_\beta^{-1}$. We build the reconstruction loss between the transformed prototypes $T_\beta^{-1}\left(G_i\left(F_1(T_\beta(\mathbf{x}))\right)\right), i = 1, \cdots, c$ and the latent representations of $\mathbf{x}_i \in \mathcal{X}_i, i = 1, \cdots, c$, respectively:

$$L_2 = \sum_{i=1}^{c} \mathbb{E}_{\mathbf{x} \in \mathcal{X}} L_{bce}\left(T_\beta^{-1}(G_i(F_1(T_\beta(\mathbf{x})))), G_i(F_1(\mathbf{x}))\right). \tag{5}$$

By integrating the equivariance property (5) with transformation $T_\beta, \beta \in \mathcal{B}$ in the interpretation regularization in (4), we propose the transformation loss as:

$$L_{trans} = \sum_{i=1}^{c} \mathbb{E}_{\mathbf{x} \in \mathcal{X}, \mathbf{x}_i \in \mathcal{X}_i} L_{bce}\left(T_\beta^{-1}(G_i(F_1(T_\beta(\mathbf{x})))), F_1(\mathbf{x}_i)\right). \tag{6}$$

Hence, we propose the objective of SITE with classification loss and transformation loss as follows:

$$\min_{G, F_1} \mathbb{E}_{\beta \sim \mathcal{B}}\Big[\mathbb{E}_{\mathbf{x} \in \mathcal{X}, \mathbf{y} \in \mathcal{Y}} L_{ce}\Big(\sigma\big(G(F_1(T_\beta(\mathbf{x})))^\top F_1(T_\beta(\mathbf{x}))\big), \mathbf{y}\Big) +$$

$$\lambda \sum_{i=1}^{c} \mathbb{E}_{\mathbf{x} \in \mathcal{X}, \mathbf{x}_i \in \mathcal{X}_i} L_{bce}\big(T_\beta^{-1}(G_i(F_1(T_\beta(\mathbf{x})))), F_1(\mathbf{x}_i)\big)\Big], \tag{7}$$

where $\lambda$ is a hyper-parameter that balances the training paces between the classification loss and the transformation loss. The first term in objective (7) ensures a transformation-aware classifier, while the second term ensures transformation-equivariant interpretations. In practice, the expectation over $\mathbf{x}_i \in \mathcal{X}_i$ and the expectation over $\mathcal{B}$ can be properly approximated by Monte Carlo sampling.

## 3.2 Visualization Methods

In the previous subsection, we obtain the self-interpretable model $G \circ F_1$, and the corresponding interpretation $\hat{\mathbf{w}}_i$ for input $\mathbf{x}$. However, since the interpretations $\hat{\mathbf{w}}_i \in \mathbb{R}^d$ are not in the original image space, the direct visualization of $\hat{\mathbf{w}}_i$ will be less meaningful.

Notice that the interpretations $\hat{\mathbf{w}}_i$ are approximations of the output space of the feature extractor $F_1$, it is natural to visualize it by visualizing $H(\hat{\mathbf{w}}_i)$, where $H : \mathbb{R}^d \to \mathbb{R}^p$ is an approximated inverse of $F_1$. And since $F_1$ is based on convolutional neural networks, a simple but judicious choice for $H$ would be the bilinear upsampling function. On the one hand, the output space of $F_1$ will preserve the relative relationship between features. And on the other hand, the Lipschitz continuity of $H$ can preserve all the intrinsic properties in $\hat{\mathbf{w}}_i$. Finally the interpretation $\hat{\mathbf{w}}_i$ is visualized in the original space of the input images by overlaying on the input $\mathbf{x}$ as a heatmap.

## 3.3 Transformation Self-Consistency Scores

In order to measure the transformation equivariance of an interpretation method properly, we propose a numerical metric, namely the *self-consistency score*. It measures the self-consistency [49] of an attribution interpretation method. For a given input data $\mathbf{x}$ and a parameterized transformation

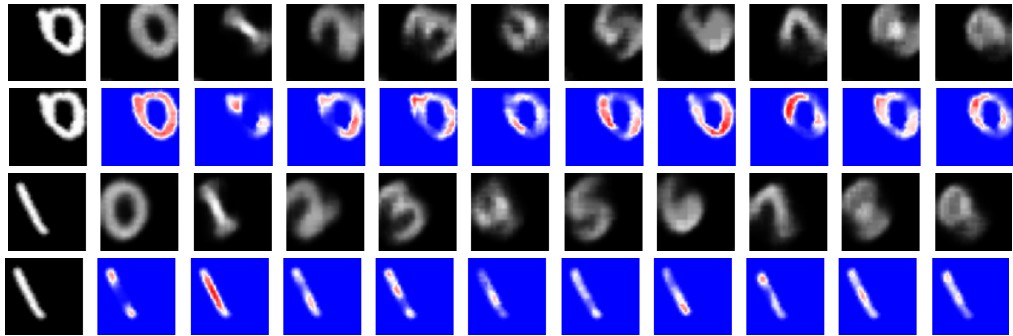

Figure 2: Interpretations of SITE on MNIST dataset. For each digit, the first column shows the randomly transformed images. The following $c = 10$ columns are the prototypes $\{\mathbf{w}_i\}_{i=1}^c$ (the first and third rows), and the interpretation heatmaps $\{\mathbf{w}_i \odot \mathbf{x}\}_{i=1}^c$ (the second and fourth rows).

$T_\beta$, let $I(\mathbf{x})$ denote the interpretation of $\mathbf{x}$, then the self-consistency score $v_\mathcal{X}(I)$ is defined as the cosine similarity between the transformation of the interpretation to $\mathbf{x}$ and the interpretation to the transformed images $T_\beta(\mathbf{x})$ as $v_\mathcal{X}(I) = \mathbb{E}_{\beta \sim \mathcal{B}} \mathbb{E}_{\mathbf{x} \in \mathcal{X}} S(T_\beta(I(\mathbf{x})), I(T_\beta(\mathbf{x})))$, where $S(\cdot, \cdot)$ is the cosine similarity. The expectation on the transformation family $\mathcal{T}_\mathcal{B}$ is approximated by the Monte Carlo sampling method. However, note that in practice $T_\beta(I(\mathbf{x}))$ transforms the interpretation directly and will introduce zero padding in the corner of interpretation heatmaps. $I(T_\beta(\mathbf{x}))$ transforms the input data before the prediction so that the interpretation is not padded. To eliminate the influence from the padded area, we introduce a transformation mask $\mathbf{m}_\beta \in \{0, 1\}^p$, where $\mathbf{m}_\beta^i = 0$ for the padding area of $T_\beta(\mathbf{x})$, and $\mathbf{m}_\beta^i = 1$ otherwise. Thus the self-consistency score is calculated by

$$\hat{v}_\mathcal{X}(I) = \mathbb{E}_{\beta \sim \mathcal{B}} \mathbb{E}_{\mathbf{x} \in \mathcal{X}} S\big(\mathbf{m}_\beta \odot T_\beta(I(\mathbf{x})), \mathbf{m}_\beta \odot I(T_\beta(\mathbf{x}))\big). \tag{8}$$

## 4 Experiments

In this section, we conduct experiments on image classification tasks with and without transformations. The experiment results demonstrate the high-quality interpretations and the validity of SITE. Please refer to the Appendix for more details about the experimental setup.

### 4.1 Experiments on MNIST

First, we implement SITE on MNIST dataset. Since SITE $G \circ F_1$ shares the same backbone structure $F_1$ with the traditional classifier $F = F_2 \circ F_1$, we clarify that SITE does not sacrifice prediction power for interpretability. Please refer to Sec. 4.3 for more details.

The interpretations of SITE on MNIST are shown in Fig. 2. The Hadamard product decides that the interpretations are essentially the pixel-wise similarities between the input digit $\mathbf{x}$ and prototype $\mathbf{w}_i$. The interpretation to each prototype can be treated as how and where do the prototype $\mathbf{w}_i$ and $\mathbf{x}$ look similar. Therefore, $\mathbf{x}$ will be classified to the class where the prototype is the most similar to the input data. Besides, we can observe that the interpretations of SITE preserve good transformation equivariance property thanks to the transformation regularization. The interpretations are transformed automatically with the input data while preserving the shape of the highlighted region.

### 4.2 Experiments on CIFAR-10

**Interpretations of** SITE

For CIFAR-10, input $\mathbf{x}$ is fed to the feature extractor $F_1$. Then the generator $G$ takes the latent representation $\mathbf{z}$ as input and generates the $c = 10$ data-dependent prototypes $\{\mathbf{w}_i\}_{i=1}^c$. Then the visualizable interpretation of the input $\mathbf{x}$ is defined by $H(\mathbf{w}_{i'} \odot \mathbf{z})$, where $H$ is the bilinear upsampling, and $i' = \arg\max_{1 \le i \le c} \mathbf{w}_i^\top \mathbf{z}$ is the predicted class for input $\mathbf{x}$.

The interpretation results are of SITE on CIFAR-10 are shown in Fig. 3. Here we sample three images of a plane, a dog, and a car for demonstrations. Each image is transformed by a constrained

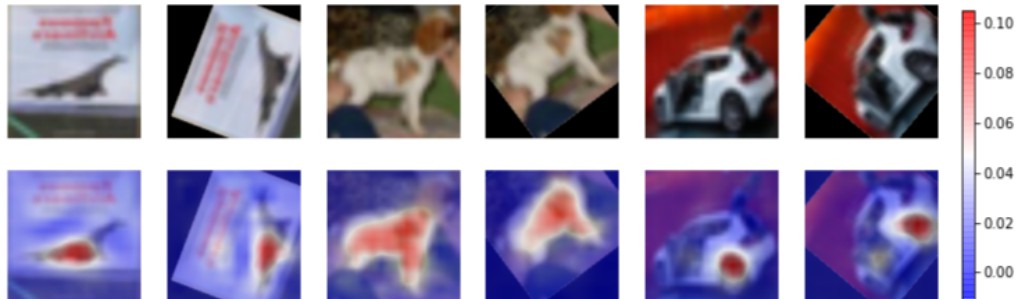

Figure 3: Interpretations of SITE on CIFAR-10 dataset. The first row shows the original images (odd columns) and their random affine transformed version (even columns). The second row shows the interpretation heatmaps overlaid on corresponding images.

affine transformation $T_\beta \in \mathcal{T}_\mathcal{B}$ that is sampled independently. And in the bottom row, we overlay the interpretations $H(\mathbf{w}_{i'} \odot \mathbf{z})$ on corresponding images. It can be found clearly that SITE successfully highlights the main parts of objects on sampled images in a comprehensible way to humans. For instance, in the dog image, SITE highlights the silhouette of the dog in both transformed and untransformed images. And comparing the odd columns and the even columns, it's clear that the interpretability of SITE preserves great self-consistency during transformations [49]. This can be treated as the robustness to transformations. Please refer to the Appendix for more examples. Besides, we would like to clarify that SITE does not sacrifice expressive power for interpretations. We take the ResNet-18 classifier as the benchmark since SITE takes the same structure as the feature extractor. Given the same transformation family $\mathcal{T}_\mathcal{B}$, SITE and ResNet-18 backbone model achieve comparable validation accuracy of 89% on randomly transformed images. And on untransformed images, SITE even demonstrate higher expressiveness. Please refer to Sec. 4.3 for more details.

**Comparison with Post-Hoc Methods**

We carry out comparison experiments with various attribution methods that interpret feature contributions to the prediction results. The comparing methods include: back-propagation methods such as Grad-CAM [41], excitation back-propagation [59], guided back-propagation [47], gradient [46], De-ConvNet [58], and linear approximation. And also there are perturbation methods such as randomized input sampling (RISE) [32] and extremal perturbation (EP) [14]. To illustrate the comparison results consistently, we use heatmaps of the same settings to visualize the interpretations of all methods. Since the interpretations of different models are obtained in very different ways, the visualizations of them are performed separately. Hence the heatmaps only demonstrate the relative importance of pixels within each interpretation itself. The comparison of the interpretation results is shown in Fig. 4, where we present the interpretations to the predictions of the given image, respectively. First, we clarify that for the sake of consistency, all post-hoc interpretations shown in Fig. 4 are obtained by applying the post-hoc interpretation models mentioned above to SITE. Since SITE is trained on the transformed dataset, all post-hoc interpretations are reasonable to the transformed image. However, their interpretations of the untransformed image are affected by the transformation. SITE does the best in capturing the main body of the ship in the untransformed image. It also preserves the best self-consistency. In fact, according to the self-consistency scores $\hat{v}_\mathcal{X}(I)$ over the whole validation set of CIFAR-10, SITE outperforms all post-hoc methods, and is thereby more robust to transformations. The comparison of self-consistency scores is shown in Table 1. Due to the inefficient computation of perturbation methods, here we omit the calculations of RISE method and extremal perturbation method. For completeness, We also include the self-consistency scores of the post-hoc methods on the backbone model (ResNet-18). It is also trained on the transformed training set.

Finally, we carry out the mask-$k$-pixels experiments [8] to demonstrate the equivariance of SITE as a self-interpretable model. This experiment is implemented by masking $k$ pixels of the input data based on the interpretations provided. For each interpretation model, we obtain a series of masked subset of the original dataset based on the interpretations. Here we sample the first 1000 images from the validation set of CIFAR-10, and perform the mask-$k$-pixels experiments on all interpretation methods mentioned above except for the two perturbation methods. Besides, we also add the case

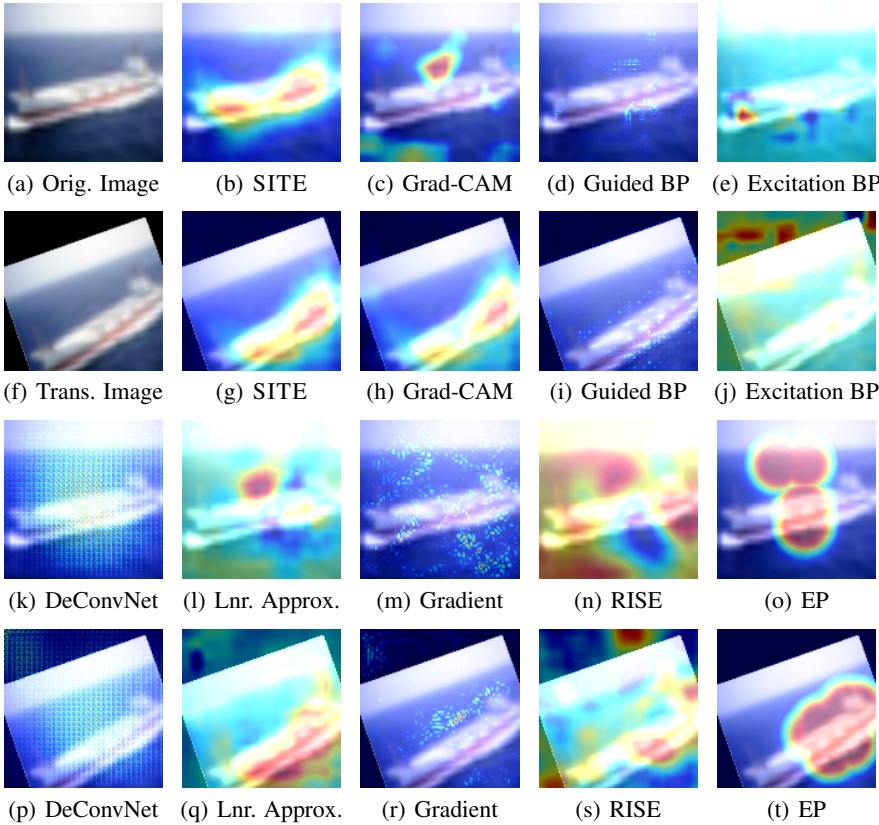

Figure 4: Interpretation comparison on CIFAR-10 dataset. The odd rows show the interpretations on the original image, while the even rows show the interpretations on the transformed image. Note that the model is trained on **transformed images**, thus all interpretations on the transformed images are relatively reasonable. However, most interpretations are highly disturbed on untransformed image, while SITE preserves the most transformation equivariant interpretations.

Table 1: Self-consistency scores $\hat{v}_{\mathcal{X}}(I) \in [-1, 1]$ of interpretation methods. A higher score indicates better self-consistency. The first row is the scores for SITE, and the second row is the scores for the backbone model (ResNet-18). Both models are trained on transformed data. Perturbation methods RISE and EP are omitted in this experiment due to the inefficient computation.

| $I$ | SITE | Grad-CAM | Guided BP | Excitation BP | Gradient | Linear Approx. | DeConvNet |
|---|---|---|---|---|---|---|---|
| $v_{\mathcal{X}}$ (SITE) | **0.8860** | 0.8817 | 0.7830 | 0.1159 | 0.7174 | 0.8485 | 0.8591 |
| $v_{\mathcal{X}}$ (backbone) | - | 0.8416 | 0.8168 | 0.2460 | 0.6926 | 0.4183 | 0.7721 |

where pixels are randomly masked. In order to demonstrate the transformation equivariance, here we mask the least $k$ important pixels according to the interpretations to untransformed images, and feed the random transformations of the masked images to the classifier. We present the trend of the prediction accuracy and the log-odds ratio (LOR) of the predicted logits to the true classes in Fig. 5(a). It is expected that the more slowly a curve drops, the better equivariance the interpretation possesses. Hence, it can be found that SITE outperforms all other post-hoc interpretation methods. Besides, we observe that when very few pixels ($< 20\%$) are masked, the decrease of SITE is almost negligibly faster than some post-hoc methods. We explain this phenomenon by presenting a typical example in Fig. 5(b). Generally, the least important pixels are located at the corners, therefore, those masks are eliminated when the corners are hidden after the transformation, as shown in the top two images in Fig. 5(b). This results in almost no mask at the beginning. As the proportion of masked pixels increase, this phenomenon is gradually alleviated, as shown in the other four images in Fig. 5(b). Furthermore, we validate the faithfulness of SITE compared with post-hoc methods on Benchmarking Attribution Methods (BAM) dataset [57]. Please refer to the Appendix for details.

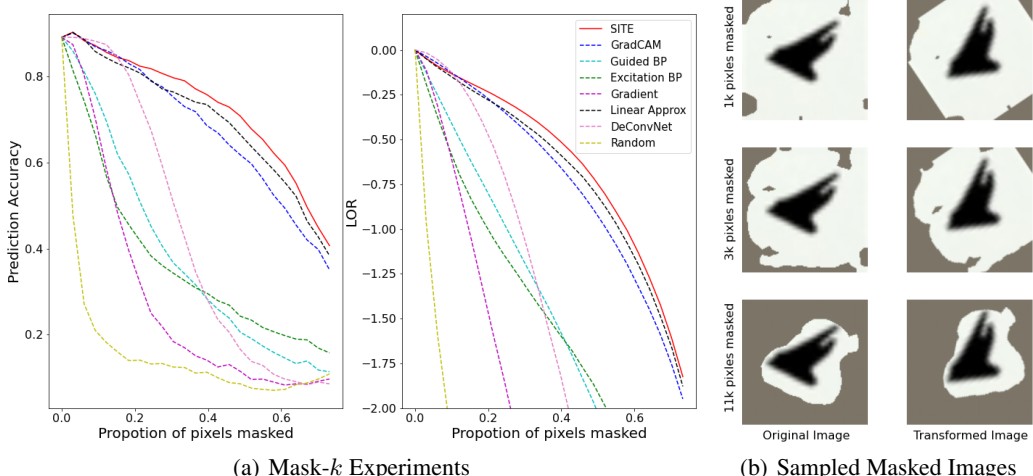

|  | (a) Mask-$k$ Experiments | (b) Sampled Masked Images |

Figure 5: (a) The trend of accuracy (left) and log-odds ratio (right) with various proportions of pixels masked; (b) Images where 1k (top), 3k (middle) and 11k (bottom) pixels (out of 16.4k pixels) are masked. The left and right columns are for the original and the transformed images, respectively.

Table 2: Accuracy comparison among self-interpretable models. We implement SITE on different backbone models and show the performance of the black-box backbone in parenthesis.

|  | Decision Tree | Random Forest | Logistic Regression | XGBoost | NAM | SITE-CNN (Blackbox CNN) | SITE-ResNet (Blackbox ResNet) |
|---|---|---|---|---|---|---|---|
| MNIST | 0.886 | 0.970 | 0.929 | 0.975 | 0.935 | **0.988** (0.981) | - |
| CIFAR-10 | 0.229 | 0.396 | 0.357 | 0.450 | 0.370 | **0.840** (0.828) | **0.892** (0.862) |

## 4.3 Expressiveness

Since there is an inevitable trade-off between expressiveness and interpretability, most existing self-interpretable models have relatively low accuracy on image datasets such as MNIST and CIFAR. Here we compare the expressiveness of SITE and existing self-interpretable models including simple models (trained using `sklearn`) like Decision Tree, Random Forest, Logistic Regression, and complex models like XGBoost [9] (trained using `xgboost`), Neural Additive Model (NAM) [2]. The results of XGBoost are reported in [33]. We include SITE with backbones of different levels of complexity to demonstrate the scalability of SITE. The CNN backbone contains 235k parameters for MNIST and 1.2m parameters for CIFAR-10, while the ResNet backbone is the ResNet-18 used in all previous experiments. The backbone models share the same structures and the same (transformed) training data as the corresponding SITE in feature extraction. The test is performed on the *untransformed* validation set. As shown in Table 2, SITE outperforms all other self-interpretable models by a large margin. It has even higher expressiveness than the backbone model. We give this credit to the regularization to the self-interpretation [5].

## 5 Conclusions

In this paper, we propose a self-interpretable model SITE with transformation-equivariant interpretations. We focus on the robustness and self-consistency of the interpretations of geometric transformations. Apart from the transformation equivariance, as a self-interpretable model, SITE has comparable expressive power as the benchmark black-box classifiers, while being able to present faithful and robust interpretations with high quality. It is worth noticing that although applied in most of the CNN visualization methods, the bilinear upsampling approximation is a rough approximation, which can only provide interpretations in the form of heatmaps (instead of pixel-wise). It remains an open question whether such interpretations can be direct to the input space (as shown in the MNIST experiments). Besides, we consider the translation and rotation transformations in our model. In future work, we will explore the robust interpretations under more complex transformations such as scaling and distortion. Moreover, we clarify that SITE is not limited to geometric transformation (that we used in the computer vision domain), and will explore SITE in other domains in future work.

## Acknowledgement

This work was partially supported by NSF IIS #1955890, Purdue's Elmore ECE Emerging Frontiers Center. The applications of various post-hoc methods are implemented through the TorchRay toolkit. We are grateful to the anonymous NeurIPS reviewers for the insightful comments.

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
