# Appendix for "Self-Interpretable Model with Transformation Equivariant Interpretation"

**Yipei Wang, Xiaoqian Wang** *
Department of Electrical and Computer Engineering
Purdue University
West Lafayette, IN 47907
`wang4865@purdue.edu, joywang@purdue.edu`

## 1 Experimental Setup

All experiments are conducted @ NVIDIA Dual RTX5000 GPUs with the Intel Xeon W-2145 CPU and NVIDIA Dual RTX6000 GPUs with the Intel i9-9960X CPU.

First, we define the family of geometric transformations to which we want SITE to be equivariant as constrained parametric affine transformations [4]: $\mathcal{T}_{\mathcal{B}} = \{T_\beta : \beta \sim \mathcal{B}\}$, where $\beta \in \mathbb{R}^6$. And $\mathcal{B}$ is defined to constrain the affine transformations to be compositions of rotations in $[-\pi/2, \pi/2]$ and translation in $[-h/2, h/2]$, where $h$ denotes the width and the length of input data.

During experiments, the model configurations can be divided into two different settings according to the complexity of the dataset. For MNIST [6], as it is simple, and the intrinsic structures are distinct by pixels among classes, the structure of SITE degenerates from $G \circ F_1$ to $G$ by setting $F_1$ to be the identical operator. That is, $\mathbf{z} = F_1(\mathbf{x}) = \mathbf{x}$. Correspondingly, the generator $G$ instead maps input $\mathbf{x}$ to its prototypes $G_i(\mathbf{x}), i = 1, \cdots, c$. Hence the structure of SITE is built to be an autoencoder-based structure, where there are $c$ parallel decoders. As for CIFAR-10 [5], due to the need for upsampling in visualization, the image data are resized to $128 \times 128$. The feature extractor $F_1$ is built based on ResNet-18 [3]. Here $F_1 : \mathbb{R}^{3 \times 128 \times 128} \to \mathbb{R}^{10 \times 16 \times 16}$. And for the generator $G$, it consists of $c = 10$ (number of categories) parallel autoencoders, such that $G_i : \mathbb{R}^{10 \times 16 \times 16} \to \mathbb{R}^{10 \times 16 \times 16}$. Both MNIST and CIFAR-10 datasets are split into the training and validation sets by default. And all presented examples are from the validation sets. We also test on more complex datasets like Food-101 [1] to demonstrate the scalability of SITE. Please refer to the Appendix 5 for details. Besides, in order to balance the classification loss and the transformation loss we set the scalar factor to be $\lambda = 5$ throughout the training phase.

## 2 More Examples on CIFAR-10

In this section, we present more results of SITE on the CIFAR-10 dataset as a supplement to *Fig. 3 in the main body of the paper*. Here we only present correctly classified examples. We sample 3 images for each class from the default validation set of CIFAR-10. The interpretations of SITE are shown in Fig.1-10. Here the first rows are the untransformed and the transformed images, while the second rows are the corresponding interpretations. And two adjacent columns are a pair of untransformed and transformed images. The results of ten classes are listed alphabetically, that is, Fig. 1 for airplanes, Fig. 2 for birds, Fig. 3 for cars, Fig. 4 for cats, Fig. 5 for deer, Fig. 6 for dogs, Fig. 7 for frogs, Fig. 8 for horses, Fig. 9 for ships, and finally Fig. 10 for trucks. It can be clearly found that SITE can accurately highlight the important features of both untransformed and transformed images in classification. Besides, SITE also demonstrates great self-consistency, as the highlighted areas preserve very similar shapes between the untransformed and transformed images.

---

*Corresponding author.

35th Conference on Neural Information Processing Systems (NeurIPS 2021), Sydney, Australia.

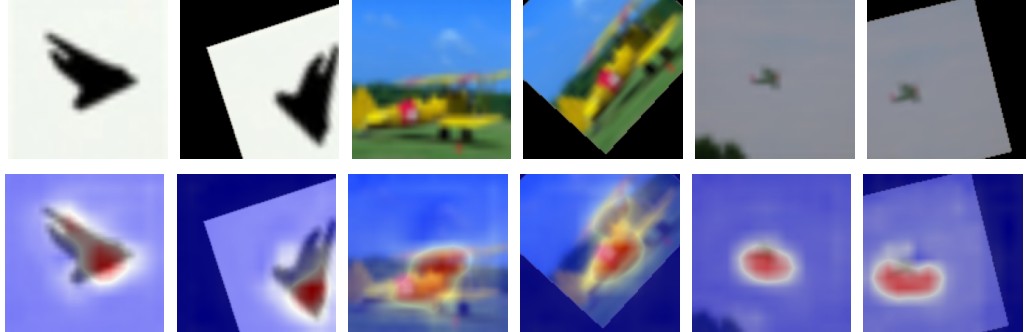

Figure 1: Additional examples of class "Airplane". The first row shows the original images, and the second row shows the heatmaps learned from SITE.

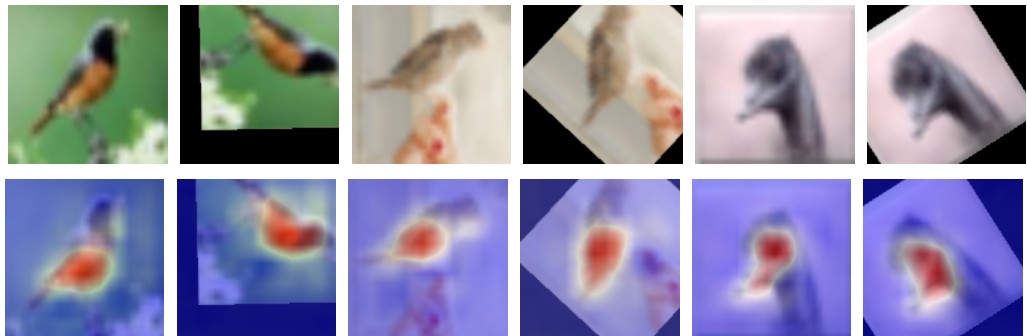

Figure 2: Additional examples of class "Bird".

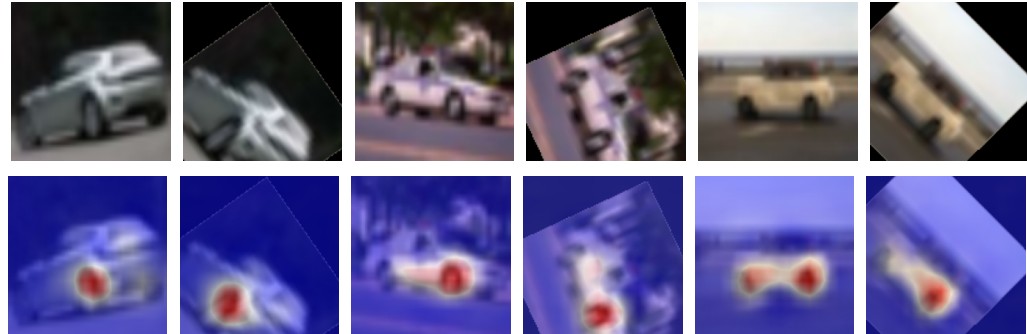

Figure 3: Additional examples of class "Car".

## 3 More Examples in Comparison

In this section, we present more comparison results among SITE and the post-hoc methods mentioned above. This is a supplement to *Fig. 4 in the main body of the paper*. The companions include: back-propagation methods such as Grad-CAM [9], excitation back-propagation [14], guided back-propagation [11], gradient [10], DeConvNet [13], and linear approximation. And also there are perturbation methods such as randomized input sampling (RISE) [8] and extremal perturbation (EP) [2]. The various comparing methods are implemented through the TorchRay toolkit [2]. To illustrate the comparison results consistently, we use heatmaps of the same settings to visualize the interpretations of all methods. Since the interpretations of different models are obtained in very different ways, the visualizations of them are performed separately. Hence the heatmaps only demonstrate the relative importance of pixels within each interpretation itself. The results are presented in Fig. 11. In Fig. 11, we demonstrate a similar result as shown in the main body. Although all methods present good results for the transformed images, post-hoc methods show less

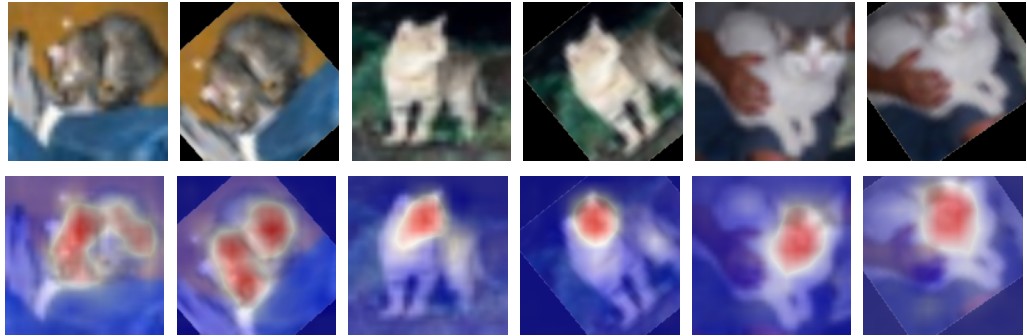

Figure 4: Additional examples of class "Cat".

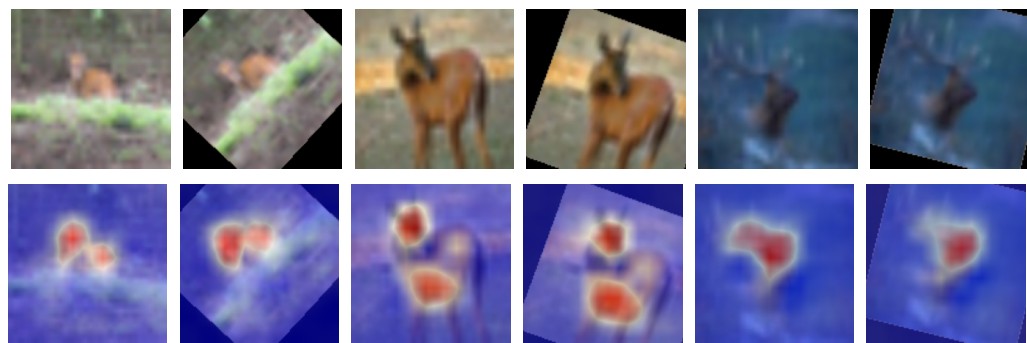

Figure 5: Additional examples of class "Deer".

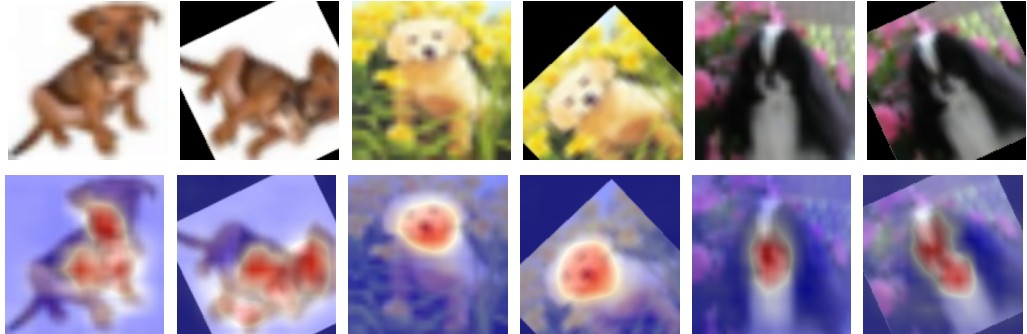

Figure 6: Additional examples of class "Dog".

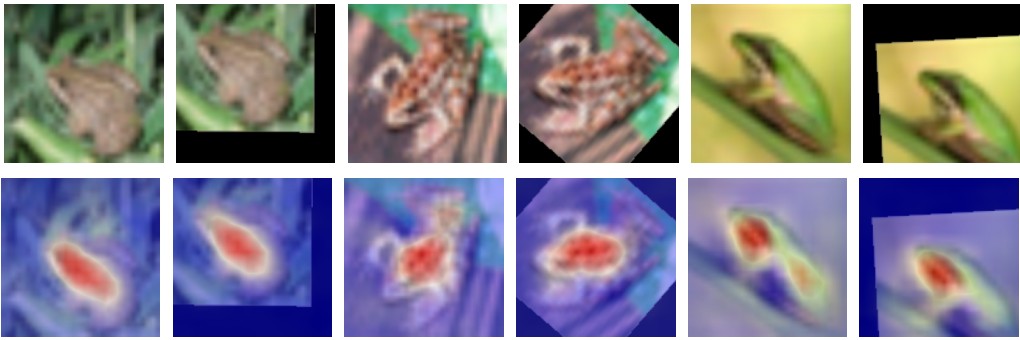

Figure 7: Additional examples of class "Frog".

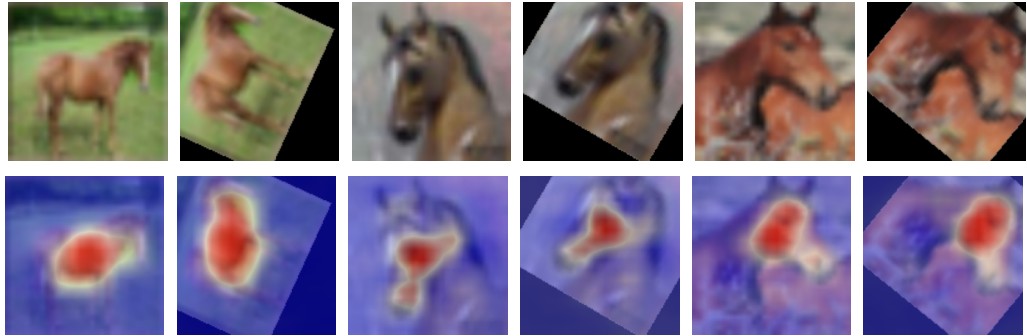

Figure 8: Additional examples of class "Horse".

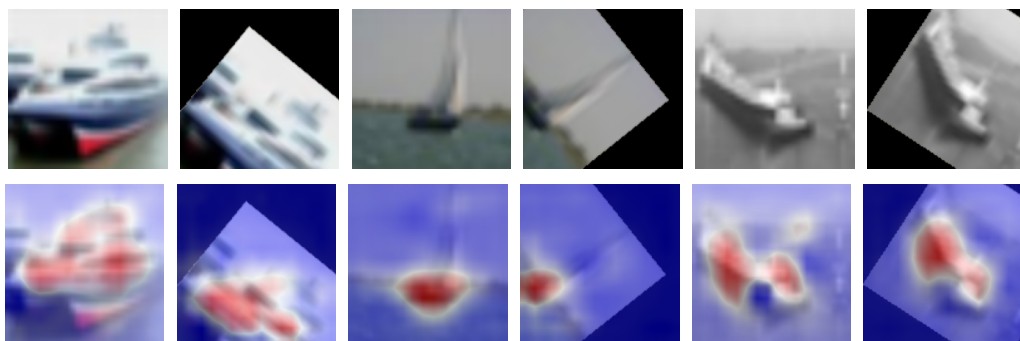

Figure 9: Additional examples of class "Ship".

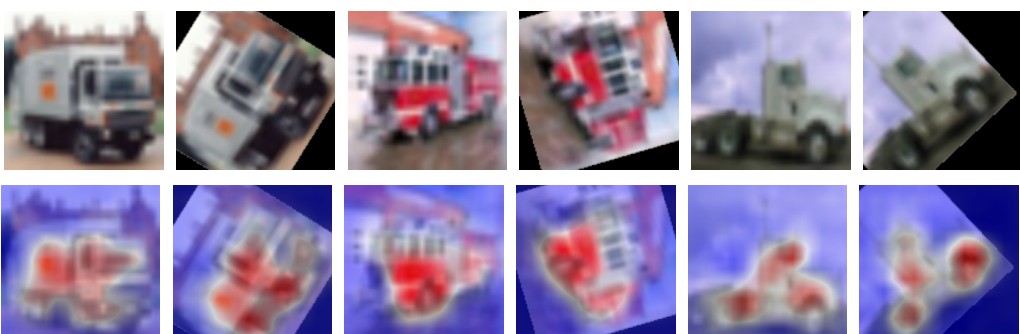

Figure 10: Additional examples of class "Truck".

faithful interpretations when dealing with untransformed images. This is because the model is trained on transformed images. And therefore, we can claim that post-hoc methods are not faithful to the predictions.

## 4 Benchmarking Attribution Methods

The Benchmarking Attribution Methods (BAM) [12] are evaluations to the correctness of attribution interpretation methods. The BAM dataset consists of artificial images, which are combinations of the scene dataset MiniPlaces [15] and objects dataset MSCOCO [7]. The dataset is constructed by overlaying the scaled objects on the scenes. Since there are 10 classes for both objects and scenes dataset, there are 100 classes of BAM dataset after the composition. And each class has 1000 images. Using the same dataset, with different labels, the BAM dataset can be denoted by $\mathcal{X}_o$ and $\mathcal{X}_s$, where $\mathcal{X}_o$ has labels for the objects, while $\mathcal{X}_s$ has labels for the scenes. An attribution interpretation method is considered to be reasonable only when it can highlight correct areas – if trained on $\mathcal{X}_o$,

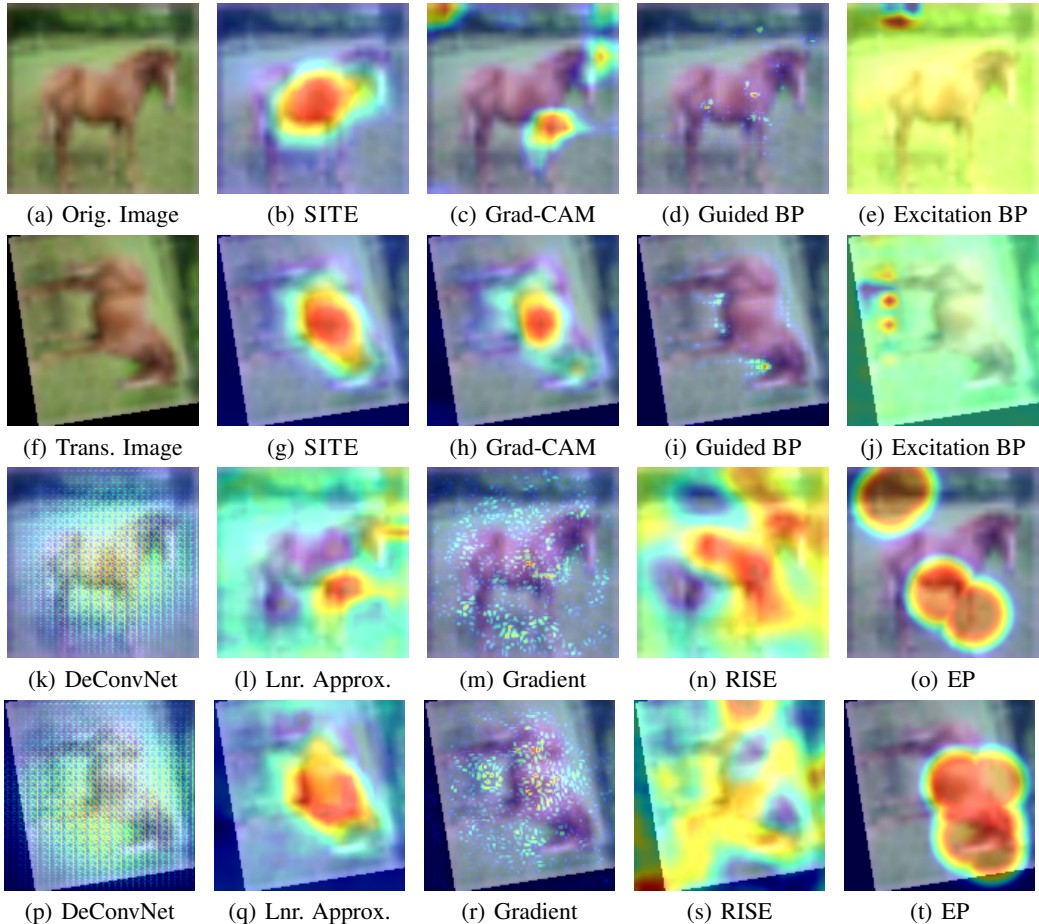

| | | | | |
|---|---|---|---|---|
| (a) Orig. Image | (b) SITE | (c) Grad-CAM | (d) Guided BP | (e) Excitation BP |
| (f) Trans. Image | (g) SITE | (h) Grad-CAM | (i) Guided BP | (j) Excitation BP |
| (k) DeConvNet | (l) Lnr. Approx. | (m) Gradient | (n) RISE | (o) EP |
| (p) DeConvNet | (q) Lnr. Approx. | (r) Gradient | (s) RISE | (t) EP |

Figure 11: Additional comparisons of interpretations from SITE and post-hoc methods. Here SITE accurately captures the important features in both transformed and untransformed images. However, most of the post-hoc methods can only have comparable results on transformed images (where the model is trained), but fail on the untransformed image.

the highlighted area should be the objects, and vice versa. We train SITE on both datasets, and present the attribution interpretations by comparing them with other post-hoc methods mentioned above. The results are shown in Fig. 12 and 13. Here Fig. 12 (a)(f) are repeated just for an aligned illustration. In Fig. 12, we present the results on an (correctly classified) untransformed image from the validation set of BAM. It can be found that most methods present reasonable interpretations of the prediction result. That is, they highlight correct areas for corresponding models (trained on $\mathcal{X}_s$ and $\mathcal{X}_o$). However, from Fig. 13, where the image is transformed, SITE shows more self-consistent and faithful interpretations than the comparing methods. Here the two images Fig. 13 (a)(f) are for object and scene datasets, respectively, and are thereby randomly transformed separately. From Fig. 13(b)(c) we can find that SITE outperforms Grad-CAM in highlighting the object area.

## 5    Results on Food-101 Dataset

Food-101 [1] is a fine-grained food image dataset. It is much more complicated than CIFAR-10 and MNIST. We use this to demonstrate the scalability of SITE. The dataset contains 101 categories and 1000 images per category. The 1000 images of each category are split into the training (750) and validation (250) subsets. Images are randomly transformed during both the training and the validation phases. We resize all images to $128 \times 128$, and use the same structure $F_1, G$ as CIFAR-10. The difference is that the number of categories is $c = 101$ instead of 10. Under this setting, $F_1$ contains around 11.2m parameters, while $G$ contains around 3.4m parameters. As a result, SITE only increases the number of parameters by 30% for such a complex dataset. The accuracy results of

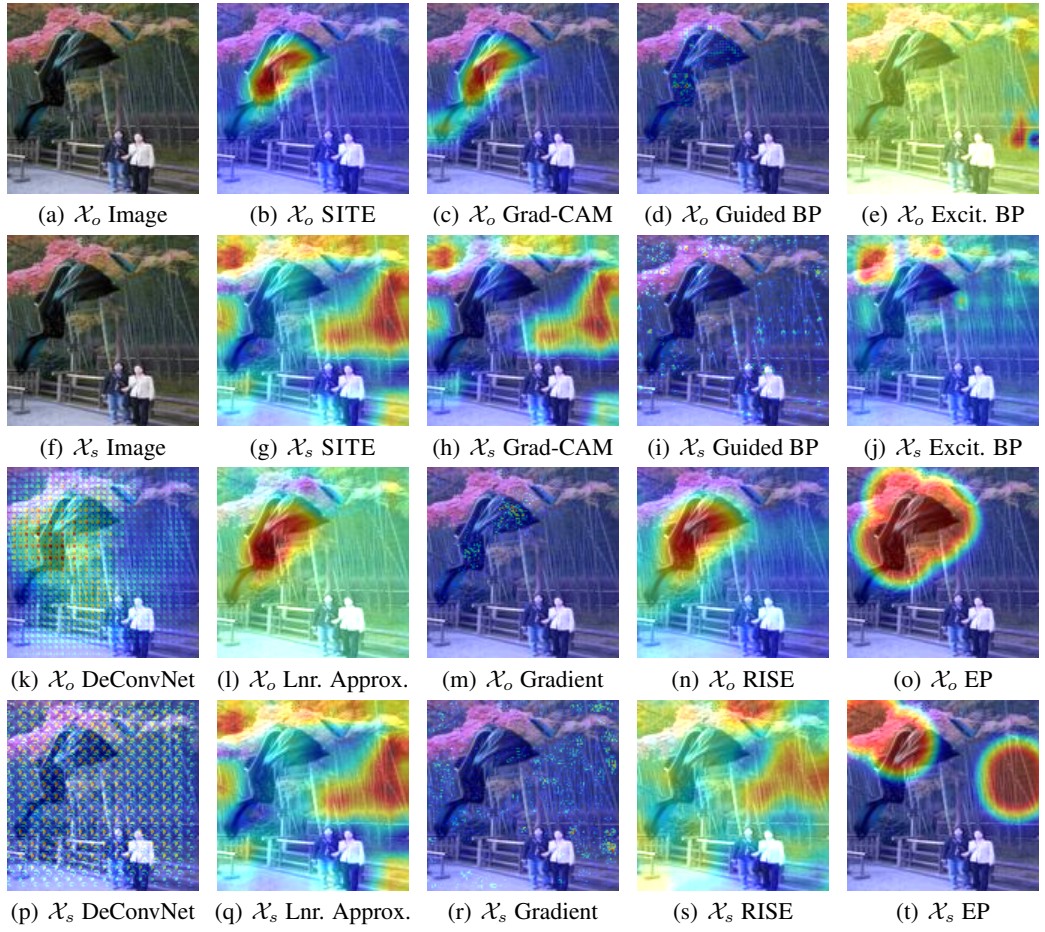

(a) $\mathcal{X}_o$ Image    (b) $\mathcal{X}_o$ SITE    (c) $\mathcal{X}_o$ Grad-CAM    (d) $\mathcal{X}_o$ Guided BP    (e) $\mathcal{X}_o$ Excit. BP

(f) $\mathcal{X}_s$ Image    (g) $\mathcal{X}_s$ SITE    (h) $\mathcal{X}_s$ Grad-CAM    (i) $\mathcal{X}_s$ Guided BP    (j) $\mathcal{X}_s$ Excit. BP

(k) $\mathcal{X}_o$ DeConvNet    (l) $\mathcal{X}_o$ Lnr. Approx.    (m) $\mathcal{X}_o$ Gradient    (n) $\mathcal{X}_o$ RISE    (o) $\mathcal{X}_o$ EP

(p) $\mathcal{X}_s$ DeConvNet    (q) $\mathcal{X}_s$ Lnr. Approx.    (r) $\mathcal{X}_s$ Gradient    (s) $\mathcal{X}_s$ RISE    (t) $\mathcal{X}_s$ EP

Figure 12: The comparison among different interpretation methods on BAM dataset. The sample is from the *untransformed* validation set. The interpretations in the first and the third rows ((a)-(e), (k)-(o)) are to the model that is trained on $\mathcal{X}_o$ (with labels for objects). And the interpretations in the second and the last rows ((g)-(h), (p)-(t)) are to the model that is trained on $\mathcal{X}_s$ (with labels for scenes).

both SITE and the backbone model (ResNet-18) are shown in Table 1. And the interpretations are illustrated in Fig. 14. SITE obviously preserves great self-consistency between transformed and untransformed images.

| Model | Accuracy |
|---|---|
| SITE | 63.91% |
| black-box backbone (ResNet-18) | 64.04% |

Table 1: The expressiveness results of SITE and the corresponding black-box model on Food-101.

## 6 Pointing Game

In order to further demonstrate the superiority of SITE in the interpretation quality compared with post-hoc models, we also carry out the pointing game experiment [14] on the annotated MNIST dataset. In the pointing game, an interpretation method calculates an interpretation map $\hat{\mathbf{w}}$ of the input $\mathbf{x}$ w.r.t. the class $c$. The method scores a hit every time the largest value of $s$ falls in the image region $\Omega$ within the tolerance $\tau$. $\Omega$ is the region containing the object for the objects dataset (or excluding

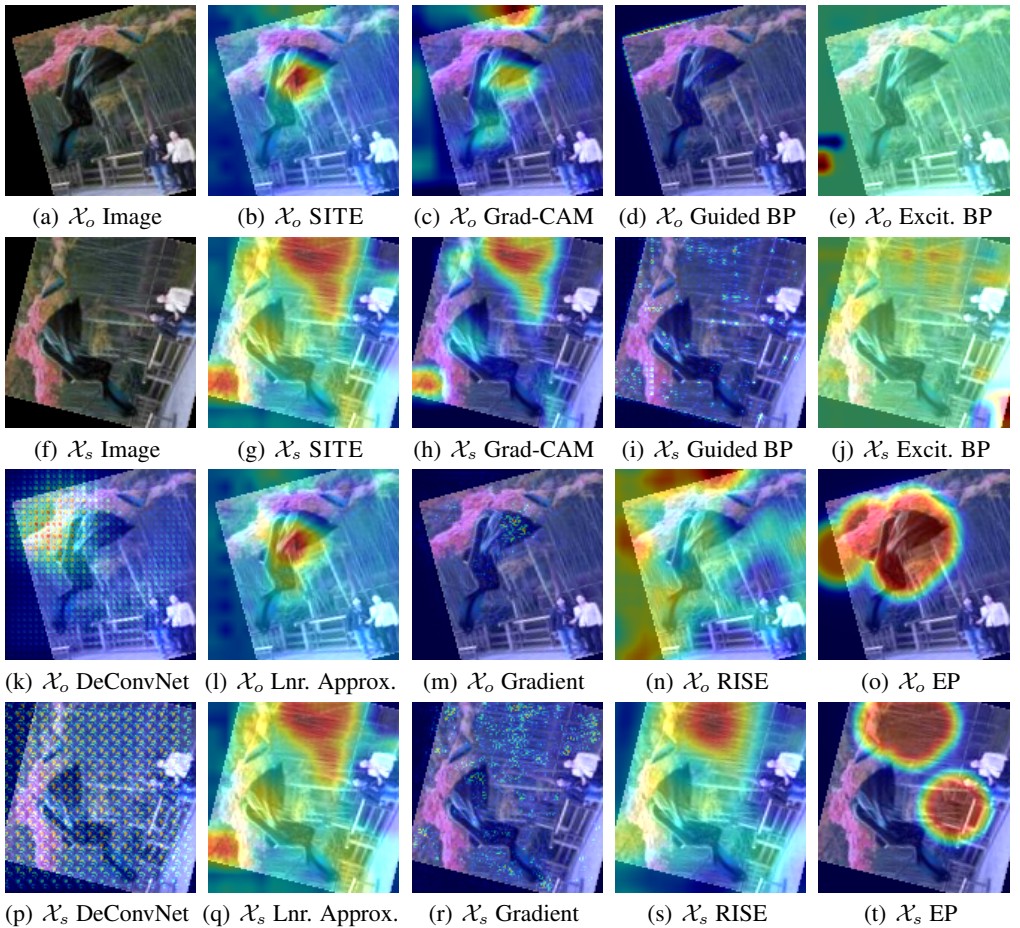

(a) $\mathcal{X}_o$ Image     (b) $\mathcal{X}_o$ SITE     (c) $\mathcal{X}_o$ Grad-CAM     (d) $\mathcal{X}_o$ Guided BP     (e) $\mathcal{X}_o$ Excit. BP

(f) $\mathcal{X}_s$ Image     (g) $\mathcal{X}_s$ SITE     (h) $\mathcal{X}_s$ Grad-CAM     (i) $\mathcal{X}_s$ Guided BP     (j) $\mathcal{X}_s$ Excit. BP

(k) $\mathcal{X}_o$ DeConvNet     (l) $\mathcal{X}_o$ Lnr. Approx.     (m) $\mathcal{X}_o$ Gradient     (n) $\mathcal{X}_o$ RISE     (o) $\mathcal{X}_o$ EP

(p) $\mathcal{X}_s$ DeConvNet     (q) $\mathcal{X}_s$ Lnr. Approx.     (r) $\mathcal{X}_s$ Gradient     (s) $\mathcal{X}_s$ RISE     (t) $\mathcal{X}_s$ EP

Figure 13: The comparison among different interpretation methods on BAM dataset. The sample is from the *transformed* validation set. The interpretations in the first and the third rows ((a)-(e), (k)-(o)) are to the model that is trained on $\mathcal{X}_o$ (with labels for objects). And the interpretations in the second and the last rows ((g)-(h), (p)-(t)) are to the model that is trained on $\mathcal{X}_s$ (with labels for scenes).

the object for the scenes dataset). The ratios $\frac{hit}{hit+miss}$ are shown in Table 2. As the baseline, we also include the post-hoc interpretations to the backbone models. It can be found that SITE outperforms all post-hoc models in the pointing game experiment. Similar to the experiments we conduct, all methods are trained with the same set of randomly transformed images (which includes the identity mapping, i.e., original images) and explain the same model. Thus, the significantly higher hit ratio by SITE in pointing game evaluation validated the improvement of SITE in transformation equivariance.

Table 2: The pointing game results on annotated MNIST dataset.

| Classifier
Interpreter | Transformed | | Untransformed | |
| --- | --- | --- | --- | --- |
| | SITE | Backbone | SITE | Backbone |
| SITE | **0.9993** | - | **0.9996** | - |
| Gradient | 0.5423 | 0.8992 | 0.7741 | 0.9540 |
| GradCAM | 0.7726 | 0.7730 | 0.8138 | 0.8062 |
| Linear Approx. | 0.6540 | 0.9577 | 0.8381 | 0.9856 |
| DeconvNet | 0.2546 | 0.6895 | 0.3128 | 0.6553 |
| Excitation BP | 0.3588 | 0.9892 | 0.4716 | 0.9979 |
| Guided BP | 0.4664 | 0.9974 | 0.8825 | 0.9990 |

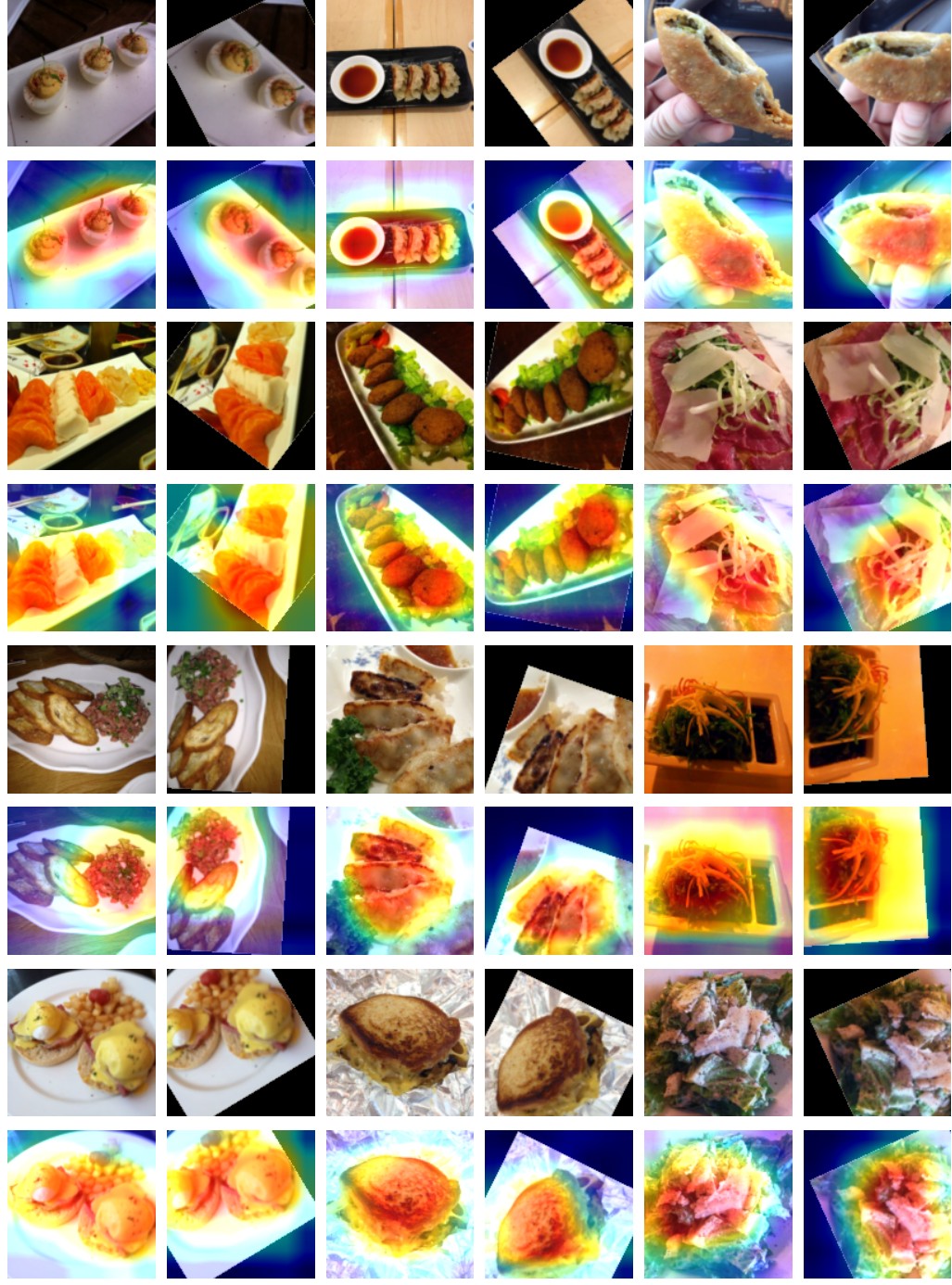

Figure 14: SITE Interpretations on Food-101

# 7 Failure Case Analysis

There's no perfect model that does not make any mistake. Therefore, the interpretations of the misclassified data, i.e. the failure cases, are very important. On the one hand, it can give users comprehensible feedback to the mistake by revealing its reasoning. On the other hand, it can help model designers to better understand and debug it, and it can also provide insights into the training data. Based on the structure of SITE, we can easily present the interpretation to the predicted logits

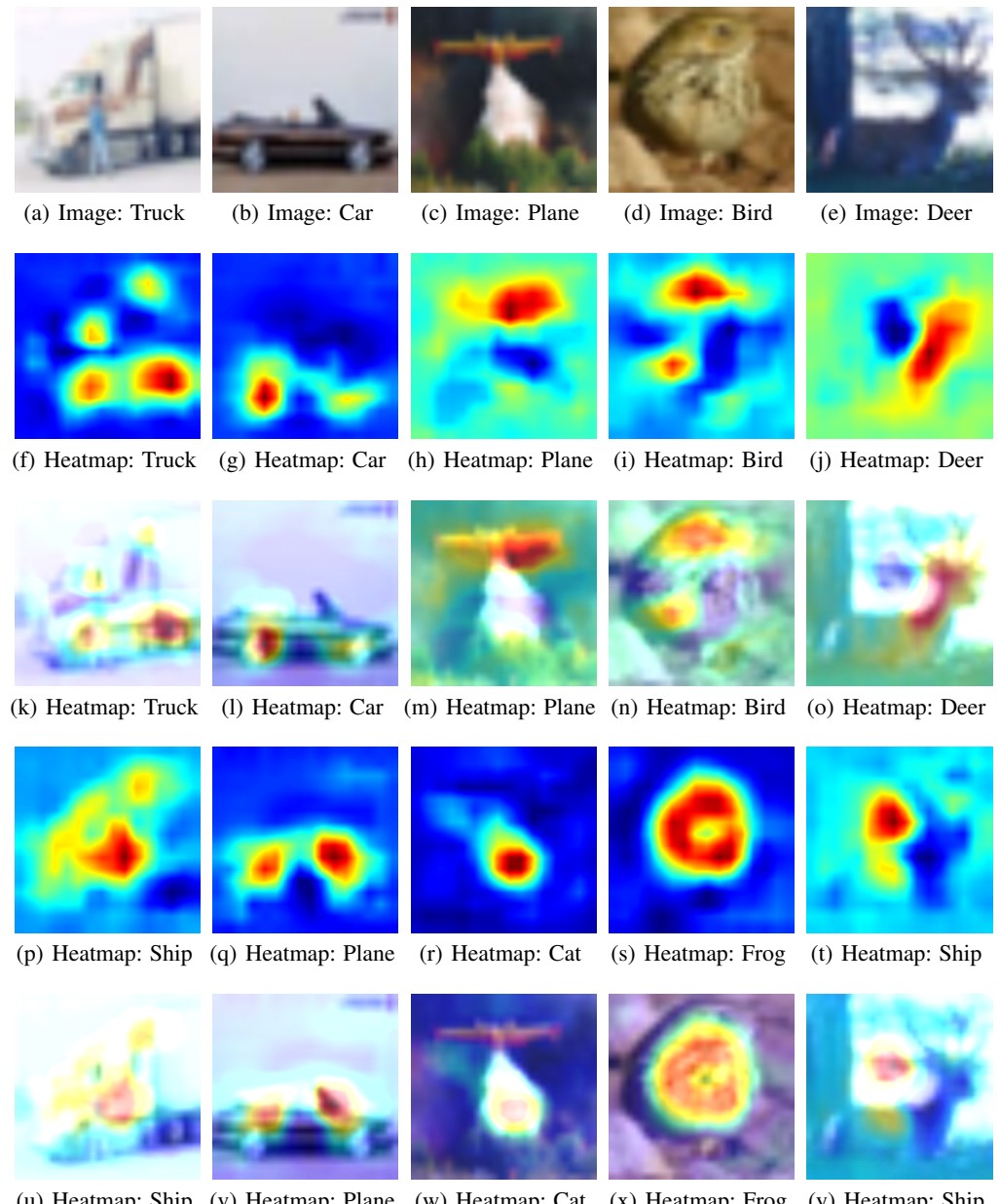

Figure 15: Failure cases of of predictions on CIFAR-10. The first row images (a)(b)(c)(d)(e) are input images in the default validation set of CIFAR-10, where SITE makes wrong predictions. The second and the third rows (f)-(o) are the interpretations SITE makes to the true classes. The second row is the heatmaps themselves, while the third row is the heatmaps overlayed on the input images. The fourth and the last rows (p)-(y) are the interpretations SITE makes to the predicted (wrong) classes. The fourth row is the heatmaps themselves, while the last row is the heatmaps overlayed on the input images. The five columns are truck, car, plane, bird, and ship, respectively. And they are classified to be ship, plane, cat, frog, and ship, respectively.

of arbitrary classes. Here in Fig. 15, we present five examples where SITE fails in predictions. For clarity concerns, we omit transformations here and use original CIFAR-10 data directly. The first row is the input images from the validation set of CIFAR-10. The second and the third rows are the interpretations of the true class, while the last two rows are the interpretations of the predicted (wrong) class.

From Fig. 15(a)(f)(k)(p)(u), we can see that a truck is predicted to be a ship by SITE. The interpretation of the prediction to the ship class is shown in (p)(u) and the interpretation to the truck class is shown in (f)(k). We can find that SITE focuses mainly on wheels for trucks but on the main body for ships.

Similarly, by comparing Fig. 15(b)(g)(l)(q)(v), we can deduce that SITE discriminates this car as a plane mainly because of the lateral view of the front windshield as shown in (q)(v). It is possible that SITE treats it as an airfoil. This can also provide an insight to the training data that there should be more lateral view of cars. And the prediction to the car class is due to the wheels of the object, as shown in (g)(l).

And in Fig. 15(c)(h)(m)(r)(w). We can see that SITE misclassifies a plane to a cat. With the interpretations shown in (r)(w), we can find this misclassification is because the white airflow fools SITE to treat it as a hairy cat.

In Fig. 15(d)(i)(n)(s)(x), a baby bird is classified to be a frog. We can find it is classified to be a frog mainly because of the main body as shown in (s)(x), while for the bird class it is mainly because of the head.

Finally, from Fig. 15(e)(j)(o)(t)(y), the image of a deer is classified to be a ship because of the horizontal ship-like structure that is behind the deer, as shown in (j)(o). And it captures the features of the deer for the deer class as shown in (t)(y).

From the above-mentioned examples, we can see that even when making wrong predictions, the faithfulness of SITE still lead to reasonable interpretations, which benefit human to understand and debug the model, and also to enhance the training dataset.