# OpenReview forum: "Self-Interpretable Model with Transformation Equivariant Interpretation"
_NeurIPS.cc/2021/Conference — NeurIPS 2021 Poster_

### Official Review · Reviewer_zd9s · 2021-07-05

**Rating:** 6
**Confidence:** 4

**Summary:**

In applications, the authors show that the model does not sacrifice its predictive accuracy compared to less interpretable models.

**Limitations And Societal Impact:**

The authors do not seem to *explicitly* address limitations and societal impact, although both come across as reasonable.

**Main Review:**

The authors introduce an interesting idea for making a self-interpretable model that can achieve reasonably good predictive performance.

The authors do a good job motivating why self-interpretable models are useful, i.e. that "self-interpretable models make interpretations at the same time of predictions, thus reveal the intrinsic mechanism of the models, and are thereby preferred by users in high-stakes tasks."

The authors then propose to follow *interpretation transformation equivariance*. This often seems quite reasonable, but the authors should note when this is a desirable property. For example, if the introduced transformations introduce artifacts into the image (e.g. large swatches of black background in the cifar images), it may no longer be reasonable to assume that the prediction mechanism should not rely on these regions.

My main criticism of the is work is that the authors should also reference (and maybe compare to) this recent paper on [self-interpretable neural networks](https://arxiv.org/abs/1806.10574) (chen et al. 2018) which use prototypes at a granular level. This papers also helps to provide localization which can provide transformation equivariant interpretation.

The author's might also want to discuss in more detail their choice to use bilinear upsampling to visualize $\hat w$ - is this the best choice? Did the authors try any other visualizations, e.g. learning an inverse function directly?

Minor points:

- there are minor typos and grammatical errors throughout the text that could be improved.
- Fig 5a xlabel has typo

**Time Spent Reviewing:**

3

---

> ### Author Response · Authors · 2021-08-10
> **The Response to the Questions**
>
> We sincerely thank the reviewer for taking the time to review our paper and provide constructive comments. In the following we address the reviewer's questions one by one.
>
> * **"The authors then propose to follow interpretation transformation ... the prediction mechanism should not rely on these regions."**
>
> We thank the reviewer for mentioning this. If the transformation introduces exogenous information that can affect the prediction, then it is reasonable that the prediction mechanism will rely on the introduced information. In these cases, additional domain knowledge may be needed to properly handle the problem.
>
> * **"My main criticism ... provide transformation equivariant interpretation."**
>
> We thank the reviewer for mentioning this. We will add more content in the literature review of self-interpretable models to the final paper.
>
> * **"The author's might also want to ... an inverse function directly?"**
>
> Thanks for the question. The bilinear upsampling possesses many good properties here. It is Lipschitz continuous, so it will preserve the relative importance from the latent space (output of CNNs) to the raw pixel space. And it is analytical and simple enough, thereby does not introduce additional black-box uncertainty to the interpretations. Therefore, bilinear upsampling is widely used in attribution methods such as GradCAM, Linear Approximation, etc. Actually, we've considered some other methods such as learning an inverse function. However, those who involve DNNs may introduce black-box uncertainties. And this is the least we want for an interpretation method. We think this is an interesting direction to explore and will leave the discussion on other visualizations in future work.
>
> * **Typos and grammatical errors**
>
> We thank the reviewer for pointing these out. We will revise them in the final paper.

---

> > ### Comment · Reviewer_zd9s · 2021-08-12
> > **Reviewer response**
> >
> > Thanks for your response - the justification for bilinear upsampling seems quite reasonable. However, my main concern is still with the comparison to the chen et al. 2018 work, so my remaining is still unchanged.

---

> > > ### Author Response · Authors · 2021-08-16
> > > **The Response to the Questions**
> > >
> > > **Thanks for your response - the justification for bilinear upsampling seems quite reasonable. However, my main concern is still with the comparison to the chen et al. 2018 work, so my remaining is still unchanged.**
> > >
> > > Thanks for your question. Although both SITE and ProtoPNet [Chen, et al. 2018] are self-interpretable models for computer vision (CV) tasks, there are several major differences between them. First, SITE generates the interpretation as a saliency map, assigning an importance value to each pixel, while ProtoPNet focuses on the similarity between the CNN output and the prototype (representative patches of the convolutional output). In this way, SITE provides a global saliency map for the entire image, while ProtoPNet focuses more on the local patches corresponding to the local areas of the image. Second, the prototype of SITE is data-dependent, while the prototype of ProtoPNet is fixed. Third, SITE ensures the transformation equivariance property for interpretation, which is yet not straightforward in ProtoPNet. Besides, ProtoPNet requires some prior knowledge or annotations to determine parameters such as the number of prototypes, while SITE does not (in SITE, the number of prototypes is the number of classes).
> > >
> > > In addition to the differences, the two methods have some things in common. Both of these methods are built upon some non-interpretable counterparts and maintain accuracy comparable to the counterparts. Both methods use the Euclidean distance ($\ell_2$-norm and cosine similarity) in the latent space to measure the similarity.
> > >
> > > We will add the above discussion to the final paper. Thanks.
> > >
> > > **Reference**
> > >
> > > [Chen, et al. 2018] "This looks like that: deep learning for interpretable image recognition." arXiv preprint arXiv:1806.10574 (2018).

---

### Official Review · Reviewer_mRVR · 2021-07-15

**Rating:** 6
**Confidence:** 3

**Summary:**

This paper proposes a self-interpretable model for images, which can produce interpretations robust to geometric transformations by learning it with transformation equivalent regularization.
The contributions of the paper are to develop the aforementioned model with high-quality faithful and robust interpretation, while preserving high expressive power, and to propose a new quantitative metric for interpretation method, called self-consistency score.

**Ethical Concerns:**

No.

**Limitations And Societal Impact:**

The authors mentioned the limitations of the study.
There is no mention about the potential negative societal impact, but I think that's OK.

**Main Review:**

### Originality

The main originality of the paper is to propose to learn a self-interpretable model to be robust to geometric transformation of images.
The essence of the proposed model is in Eq. (7), and the proposed model simply uses the geometric transformation introduced in [16] in the equation,
The formulation of the proposed model may be new, but it seems that the formulation is a straightforward combination of the existing techniques.

### Quality

I think the proposed model is technically sound, except for some points stated in Clarity section.
Also, the experimental result shows that the proposed model can produce appropriate interpretations for geometrically transformed images as expected.

However, I'm concerning about the superiority of the proposed model compared to the post-hoc interpretation methods.
I think that if the backbone network was trained on data with data augmentation including geometric transformation, the post-hoc interpretation methods can produce interpretations as well as the proposed model.
However, Figure 4 shows that some of the comparing methods are sensitive to the geometric transformation.
Is the backbone network used in the comparing post-hoc interpretation methods the same as that of the proposed model?
If this is true, was the backbone network trained on data with data augmentation including geometric transformation?

In addition, I have another concern about the computational efficiency of the proposed model.
In Eq (7), it seems that the computational cost of the second term is very high.
The authors say "In practice, the expectation over xi \in Xi and the expectation over B can be properly approximated by Mont Carlo sampling.", but I could not find the explanation of how many samples need to approximate the expectation.
In the experiment, how many samples are used in the computation of the expectation? Also, did you investigate the tradeoff between the number of the samples and performances such as self-consistency score and accuracy?

### Clarity

The paper is well-organized and easy to follow.
However, I think the following points are unclear.

In the second term of Eq. (7), why is binary cross entropy used for measuring discrepancy of two continuous vectors in the feature space?
Intuitively, I think that Euclid distance is the first choice for measuring the discrepancy.

For transformation $T_\beta$, $T_\beta^{-1}$ is not exactly inverse of $T_\beta$, because $T_\beta$ transforms a point to another point in the original input space, while $T_\beta^{-1}$ transforms a point in $\mathbb{R}^d$ to a point in the space of $\mathbf{z}$.
The authors should state the difference between $T_\beta$ and $T_\beta^{-1}$.

In Section 3.2, the authors state that the interpretations can be visualized by bilinear upsampling.
This seems that $F_1$ is assumed to be the function that return the output of the convolution layer in CNN.
When extracting features from images using CNN, one often uses the output of the fully-connected layer in the CNN.
To avoid misreading, I think the authors should clearly state which layers in the CNN can be used for $F_1$.

### Significance

In the experiment, it seems that the proposed model achieves better accuracy and interpretability.
However, I think that the comparison with the existing models in the following perspectives is not enough.

Is the backbone network used for the comparing post-hoc interpretation methods trained with enough data augmentation including geometric transformation?

Why was not the proposed model compared with attention-based methods, such as class activation map (CAM)? Since both the proposed model and the attention-based methods do prediction and interpretation in a single model, I think that the authors should compare the proposed model with at least one attention-based method in the experiment.

**Time Spent Reviewing:**

8h

---

> ### Author Response · Authors · 2021-08-10
> **The Response to the Questions**
>
> We sincerely thank the reviewer for taking the time to review our paper and provide constructive comments. In the following we address the reviewer's questions one by one.
>
> * **"However, I'm concerning about the superiority of the proposed model ... including geometric transformation?"**
>
>
> We thank the reviewer to give us an opportunity for clarification. We would like to first clarify two different properties: 1) a classifier with robust predictions to transformation; 2) an interpretation with transformation equivariance property. The first refers to that a classifier can correctly classify the image before/after the image is transformed. While the latter refers to that the interpretation method provides equivariant interpretations before/after the transformation.
>
> For the first property, all models (SITE, backbones, etc.) are trained on the same set of randomly transformed dataset (which includes the identity mapping, i.e., original images), and thereby can be treated as robust to transformations. Thus, it is reasonable that both SITE and backbones can make accurate prediction that is robust to transformation.
>
> For the second property, however, even though the classification prediction is correct for images before/after transformations, the post-hoc interpretation methods did not provide interpretations that are robust to transformations (shown both in Table 1 and Figure 4), which is the issue that we addressed in SITE. In SITE, we provide a self-interpretable classifier with both two properties.
>
> Besides, we would like to clarify that for the sake of controlling variates, the post-hoc interpretations in Figure 4 are the interpretations to the prediction of SITE instead of the backbone network. Because all interpretation methods should be explaining the same model. And as a complement, we also include the self-consistency score of post-hoc interpretations to the backbone model (Table 1, row 2). It can be found that SITE still outperforms all post-hoc methods.
>
> Thanks for the question. We will make this claim clearer in the final paper to avoid possible ambiguities.
>
>
> * **In addition, I have another concern about ... such as self-consistency score and accuracy?**
>
> Since the sampling of the transformation parameters $\beta$, the reconstruction target $\mathbf{x}_i$, and the training data $\mathbf{x}$ are **independent**, we sample them jointly  for the sake of efficiency. Throughout the training process, we sample a batch of $\mathbf{x}_i$ from $\mathcal{X}_i$ and a batch of $\beta$ from $\mathcal{B}$ for each batch of data $\mathbf{x}$. Therefore, the additional computational burden is acceptable.
>
> * **In the second term of Eq. (7) ... for measuring the discrepancy.**
>
> When both the label and the reconstruction are between $[0,1]$, the binary cross entropy serves the same goal as Euclidean distance. They are interchangeable for tasks like autoencoders. Here we prefer the binary cross entropy loss to Euclidean distance because when updating the networks, it is more compatible with the cross entropy loss used in the first term of Eq. (7) for the classification tasks. In this way, the two terms are easier to balance in practice.
>
> * **For transformation $T_\beta$, $T_\beta^{-1}$ ... difference between $T_\beta$ and $T_\beta^{-1}$.**
>
> Defined by the parameters $\beta\in\mathbb{R}^6$, the affine transformation $T_\beta$ is actually a coordinates transformation operator. That is, it can take input of any size and transform it. We will clarify this in the final paper.
>
> * **In Section 3.2, the authors state ... in the CNN can be used for.**
>
> $F_1$ doesn't contain fully connected layers. In our experiment, $F_1$ contains all convolutional layers but no fully connected layer of $F$. We will clarify this in the final paper.
>
> * **Is the backbone network used for the comparing post-hoc interpretation methods trained with enough data augmentation including geometric transformation?**
>
> Yes. Both the backbone network and SITE are trained with the same set of randomly transformed images (including identity mapping, i.e., original images).
>
> * **Why was not the proposed model ... method in the experiment.**
>
> We would like to clarify that CAM is a special case of Grad-CAM, which is already included in the experiments. And to our best knowledge, most existing attention-based methods for CV tasks all focus on high-level features, such as [Fu et al. 2017], [Dosovitskiy et al. 2020], etc. In contrast, SITE presents pixel-wise interpretations. This main difference makes the quantitative comparison infeasible. We will explore more attention-based methods and seek similarities and possible comparisons.
>
> **Reference**
>
> [Fu et al. 2017] Fu, J., Zheng, H., & Mei, T. (2017). Look closer to see better: Recurrent attention convolutional neural network for fine-grained image recognition. In Proceedings of the IEEE conference on computer vision and pattern recognition (pp. 4438-4446).
>
> [Dosovitskiy et al. 2020] Dosovitskiy, A., Beyer, L., Kolesnikov, A., Weissenborn, D., Zhai, X., Unterthiner, T., ... & Houlsby, N. (2020). An image is worth 16x16 words: Transformers for image recognition at scale. arXiv preprint arXiv:2010.11929.

---

> > ### Comment · Reviewer_mRVR · 2021-08-30
> > **Reviewer Response**
> >
> > Thank you for the responses.
> > Since my concerns have been dispelled by the authors’ explanation, I will change my score from 5 to 6.

---

> > > ### Author Response · Authors · 2021-08-30
> > > **Thank You**
> > >
> > > Thank you again for taking the time to review our paper and provide valuable feedback! We will revise our final paper based on the reviewer’s comments.

---

### Official Review · Reviewer_FoQp · 2021-07-16

**Rating:** 6
**Confidence:** 4

**Summary:**

This paper presents an approach that learns a 'self-interpretable' model, which produces explanations that are invariant to certain user defined transformations. The attribution maps of traditionally trained models change when certain transformations (like rotation etc) are applied to the input, which could be seen as a disadvantage of these methods. Here the paper counteracts this by learning models that are self-interpreting (via higher level prototypes) and whose attribution maps are invariant to user-defined attributions.

**Ethical Concerns:**

I don't see any ethical concerns with this work.

**Limitations And Societal Impact:**

The authors should include a more explicit limitations section in their paper.

**Main Review:**

### Originality
This paper tackles an original problem within the model/dnn interpretability literature. Self-explaining models have been proposed in the past, however, none of that previous work has emphasized equivariance, which is the primary focus of this work. The paper also proposes a self-consistency score for explanations. As far as I am aware, this score is novel and could be used in future work.

### Quality
The paper tackles an important problem. However, I think the paper conflates two problems at once. First, the need for explanations to be robust to transformations is important; however, this only matters *if* the model in question is robust to such transformations. If a model is not robust to a transformation, then we shouldn't expect explanations from that model to also be robust to transformations well. The formulation proposed should also be compared to self-interpretable variants instead of just post hoc explanations alone.

### Clarity
The paper is relatively easy to follow and free of major typos, but certain references could have been made more clear in different places. I"ll go into detail on this at the end of this review.

### Significance
The insight that attributions are not transformation invariant is significant and one that I believe future work will address as well. Equivariant self-interpretable networks is also an important and interesting direction as well. As it stands these two ideas are somewhat muddled in this work, but it could be that with better presentation the message of this work would be quite significant. Figure 3 and 4 are quite nice illustrations of the benefits of the proposed approach in practice.

### Additional Comments/Questions
1) The self interpretable claim. This paper claims that the prototypes that are output by the presented network are self-interpretable, but it does not test or justify this claim. While other papers might have done this as well, it is not clear that if the prototypes derived in this work were presented to an end-user they would find it interpretable.

2) Why should we expect transformation invariant attribution from models that are not themselves transformation invariant? We now know that if one rotates an image and such image is presented to a network that has not learned such rotation transformation, then the prediction of the model can change as well. For such models one would expect the attributions to also not be rotation invariant. In the caption of figure 4. it says that the model is trained on transformed images ; however, does the prediction of the model stay the same even when these transformations are applied to the test image(s)?

3) Comparison to attribution methods. While I understand that the attribution methods that the authors compare to allow them to demonstrate the effectiveness of their method, I think self-interpretable methods are the better comparisons here. See the work: "This looks like that" by Chen et. al. and Concept Bottleneck Models by Koh et. al. These approaches should form better comparisons for this work in addition to the post hoc methods.

4) Interpreting the self-consistency score. The idea of this score is interesting; however, based on table 1 it is a bit difficult to understand or contextualize what these scores should mean. Excitation BP seems quite low, but the rest are above 0.5...does that mean that these methods are reliable? Figure(s) 3 and 4 say otherwise, so could it be that the score doesn't not really help distinguish reliable from unreliable methods?

5) Section 3.1 stops without going into detail about how the model is actually trained. This information is important even if it is with sgd type methods with standard autodiff. This would help with future replication.

#### Minor typos & Writing Feedback
Line 193: Mont Carlo -> Monte Carlo
Line 177: instead of referencing [16], it would've been better to discuss or mention the specific transformations you are referring to here.

**Time Spent Reviewing:**

6.5

---

> ### Author Response · Authors · 2021-08-10
> **The Response to the Questions**
>
> We sincerely thank the reviewer for taking the time to review our paper and provide constructive comments. In the following we address the reviewer's questions one by one.
>
> * **"The paper tackles an important problem. However, I think ... also be robust to transformations well."**
>
> We thank the reviewer to give us an opportunity for clarification. We would like to first clarify two different properties: 1) a classifier with robust predictions to transformation; 2) an interpretation with transformation equivariance property. The first refers to that a classifier can correctly classify the image before/after the image is transformed. While the latter refers to that the interpretation method provides equivariant interpretations before/after the transformation.
>
> For the first property, all models (SITE, backbones, etc.) are trained on the same set of the randomly transformed dataset (which includes the identity mapping, i.e., original images), and thereby can be treated as robust to transformations. Thus, it is reasonable that both SITE and backbones can make accurate predictions that is robust to transformation.
>
> For the second property, however, even though the classification prediction is correct for images before/after transformations, the post-hoc interpretation methods did not provide interpretations that are robust to transformations, which is the issue that we addressed in SITE.
>
> In SITE, we provide a self-interpretable classifier with both two properties. We will make this claim clearer in the manuscript to avoid possible ambiguities.
>
>
> * **"The self interpretable claim. This paper claims ... they would find it interpretable."**
>
> We demonstrate the self-interpretability in lines 159-168 of the manuscript. A model is called self-interpretable if it can make the prediction present the corresponding interpretation in the same stage, and the interpretation can reveal the true mechanism of the prediction. Inspired by linear models, which are axiomatically treated as self-interpretable, the interpretations of SITE are directly involved in the prediction process, too. Therefore, SITE is justified as self-interpretable.
>
>
>
> * **Why should we expect transformation invariant attribution from models that are not themselves transformation invariant? ... applied to the test image(s)?**
>
> Please refer to our answer to the first question. We would also like to clarify that both the SITE models and the backbone models (all trained on transformed images) show full expressiveness on both transformed and untransformed validation sets. This is because identity mapping is included in the random transformations in the training process. The accuracy of the untransformed validation set can be found in Table 2. And the accuracy of SITE on transformed set is at the same level as or even higher than those of untransformed set. The details of transformed validation accuracy will be included in the manuscript.
>
>
> * **The formulation proposed should also be compared to self-interpretable variants instead of just post hoc explanations alone.**
> * **"Comparison to attribution methods. While I understand ... in addition to the post hoc methods."**
>
> In this question, the first line is from the Quality section of the review, and the second line is from the Additional Comments/Questions. We combine the answer to the questions here.
>
> We thank the reviewer for mentioning these self-interpretable methods. We will include them in the literature review of the manuscript. As for the comparison, SITE generates interpretations as a saliency map, which assigns an importance value to each pixel. However, Chen et al. focus on the similarities between prototypes (which are representative patches of convolutional output), and Koh et al. focus on learned high-level concepts. Neither of them can be properly compared with SITE. And this is the reason we mainly compare with post-hoc methods - most of them are pixel-wise. Besides, Neural Additive Model (NAM) is one of the most recent self-interpretable neural networks with pixel-wise interpretations. And SITE outperforms NAM by a large margin.
>
>
> * **Interpreting the self-consistency score ... from unreliable methods?**
>
> We thank the reviewer for mentioning this. From Figure 4(e)(j), it can be found that Excitation Back-propagation (EBP) has indeed the worst performance on CIFAR dataset. And we also double check the self-consistency score since the score of EBP is too low. The reason is that we used to calculate the self-consistency score batch-wise. However, compared with other methods, the EBP saliency maps of different instances can be very different in scale, resulting in very low value when calculated batch-wise. Therefore, we re-calculate the self-consistency scores instance-wise and add the results below. Please compare it with Table 1 in the main paper. We can observe that score of EBP is still the lowest, but at the same scale as others. It can be found that the calculation only affects the EBP method, and has an insignificant influence on others.
>
> $I$		| SITE 		| GradCAM	| GBP		| EBP		| Gradient	| LA	| DeConvNet |
> -------|-----------|-----------|-----------|-----------|-----------|-------|-----------|
> SITE 	| **0.8860**| 0.8817	| 0.7830	| 0.1159	| 0.7174	| 0.8485| 0.8591	|
> Backbone| - 		| 0.8416	| 0.8169	| 0.2460	| 0.6926 	| 0.8485| 0.8591	|
>
> * **Section 3.1 stops ... future replication.**
>
> Thanks for the suggestion. The SITE model is trained with standard criteria using Adam optimizer. We will include more implementation details in the manuscript. We are ready to release our code and make it publicly available when the paper is published.
>
> * **Minor typos and writing feedback**
>
> We thank the reviewer for the suggestions. We will revise them in the final paper.

---

> > ### Comment · Reviewer_FoQp · 2021-08-26
> > **Thank you for the clarifications**
> >
> > I have read the author response and it does, indeed, clarify several issues that I had with the work. I agree with the authors that since their work provides explanations in terms of saliency maps, it unclear how to directly compare to prototype methods like the work of Chen et. al. However, with this all said, I think it is still important to be cautious with the term self-interpretable. One can train a 1 million parameter linear model. The coefficients of the model constitute a form of explanation and feature importance, yet, the entire 1 million parameter model might not be 'interpretable' to an end-user.
> >
> > Self-interpretable seems to be defined in this work as composable based on 'easy to derive' parts. I agree that the constituent parts can form an explanation/interpretation, however, I disagree with the claim that a model/explanation can be assessed to be self-interpretable without an end-user.
> >
> > This is overall a minor point, but it is important to soften the 'interpretability' claim since that part of this work isn't really evaluated in any sense.

---

> > > ### Author Response · Authors · 2021-08-29
> > > **Thanks for the Review**
> > >
> > > We are very grateful to the reviewer for recognizing our work and pointing out the importance of end users in explanation/interpretation. We acknowledge that this is a very interesting and practical topic. We will carefully revise the wording in our claim and avoid possible ambiguity.

---

### Official Review · Reviewer_XcMQ · 2021-07-19

**Rating:** 6
**Confidence:** 3

**Summary:**

The paper proposes SITE Self-Interpretable model with Transformation Equivariant Interpretation.

Features are first extracted by feature extraction network such as CNN/Resnet F(x) such that F(x)=z SITE uses generative model G that maps z to c prototypes each corresponds to a specific class. y_hat = softmax(G(z)^T x) where y_hat_i = softmax(w_i^Tz) so  w_i is the interpretation of the i-th prediction. The paper then proposes to to regularize the prototypes for an input image x, they enforce each generated prototype to be similar to its corresponding class’s latent representation. They also propose to regularize on the transformation equivariance property of interpretation produced by the model.

The paper also proposed self-consistency score to compare different interpretation methods. For an input x and transformation T and interpretation I(x) the self-consistency score is defined as the cosine similarity between the transformation of the interpretation to x and the interpretation of the transformed images cosine similarity(T(I(x)),I(T(x)) . Higher indicates a more robust interpretation.

The paper compared SITE interpretation with post hoc interpretability methods and the accuracy maintained by SITE with that of inherently interpretable architecture that exhibit a decrease in accuracy due to their interpretable nature. Comparison was done on MNIST and CIFAR10.

**Main Review:**

Strength:
- The use of transformation for interpretation is original and novel.
- There is a clear advance in terms of interpretation robustness when comparing SITE to post hoc methods.
- In comparison to inherently interpretable architectures SITE was able to maintain model accuracy while providing reasonable interpretation.
- The paper is well written and clear.
- The paper is technically sound.
- The proposed Self-consistency score can be used to compare different methods in the future.

Weakness:
- SITE was not compared with other self explaining networks such as SENN [1], INVASE [2], FRESH [3].
- Experiments were done on pretty basic datasets MNIST, CIFAR; will SITE be able to provide good explanations while maintaining accuracy for on more realistic datasets like imagenet, food101, and bird260.
- SITE does not provide feature-based explanations which are often desired, this makes applying SITE on other domains such as language and time-series not straightforward.
- Computational cost of SITE compared to training a blackbox Resnet was not discussed.
- Masking and measuring accuracy drop can result in false results since the accuracy drop might be the result of the masking not the removing of informative/non-informative features[4].
- The relationship to previous work in Self-Interpretable Models is not clearly stated.

[1] David Alvarez-Melis and Tommi Jaakkola. Towards robust interpretability with self-explaining neural networks. In Advances in Neural Information Processing Systems (NeurIPS), pages 7775–7784, 2018

[2] Yoon, Jinsung, James Jordon, and Mihaela van der Schaar. "INVASE: Instance-wise variable selection using neural networks." International Conference on Learning Representations. 2018.

[3] Jain, Sarthak, et al. "Learning to faithfully rationalize by construction." arXiv preprint arXiv:2005.00115 (2020).

[4] Sara Hooker, Dumitru Erhan, Pieter-Jan Kindermans, and Been Kim. A benchmark for interpretability methods in deep neural networks. In Advances in Neural Information Processing Systems, 2019.

The main reason behind my score is the limited empirical results only very simple datasets were used and other self explanatory networks were not compared.






################# Post rebuttal
Thank you for addressing my concerns. Given the author response I am happy to increase my score 5->6

**Time Spent Reviewing:**

3

---

> ### Author Response · Authors · 2021-08-10
> **The Response to the Questions**
>
> We sincerely thank the reviewer for taking the time to review our paper and provide constructive comments. In the following we address the reviewer's questions one by one.
>
> * **SITE was not compared with other self explaining networks such as SENN, INVASE, FRESH.**
>
> We attach great importance to the comparisons with existing self-interpretable models. However, it should be noticed that SITE presents interpretations as a saliency map, which is in the pixel level. Therefore, as shown in Table 2 in the main paper, we include self-interpretable models that present interpretations in this way, too. As for the three mentioned methods, SENN uses a feature extractor, and generates interpretations in that high-level space instead of the raw pixel space. FRESH is an interpretation model specifically for natural language processing (NLP) tasks instead of computer vision (CV) tasks. And although INVASE can be implemented in the pixel space, it was implemented in the $16\times 16$ pixel-patch space for CV tasks in the original paper.
>
> We add new comparison experiments with INVASE on the transformed MNIST dataset pixel-wise. INVASE has two classifiers, which are predictor and baseline. For fair comparison, both classifiers in INVASE have the same structures as the backbone network of SITE, and the selector in INVASE is a CNN-based autoencoder. Both INVASE and SITE are trained on the same set of randomly transformed images (which includes the identity mapping, i.e., original images). The results are summarized as follows:
>
> Model 				| Accuracy
> ----------------------|---------
> INVASE - baseline 	| 94.7%
> INVASE - predictor	| 91.7%
> SITE 					| 98.8%
>
> We can notice that the SITE shows better accuracy than the two classifiers in INVASE, which support the expressive power of SITE over comparing self-interpretable models. Besides, INVASE is after all a feature-selection method, which is essentially different from interpretation models. Here the predictor and the baseline models have completely different parameters. The selected features are not compatible with the baseline network, while the original images (full set of features) are not compatible with the predictor network. Therefore, the selected features of INVASE cannot be used as interpretations to either network.
>
>
> * **Experiments on more realistic datasets like imagenet, food101, and bird260.**
>
> We present only the CIFAR and MNIST results because these two image datasets are already complex enough to existing pixel-level self-interpretable models (as shown in Table 2 in main paper). SITE can be easily scaled up by increasing the complexity of the backbone model. Due to the time limit of rebuttal, we add a new experiment on the transformed food101 dataset with the structure of ResNet-18. The accuracy on the transformed validation dataset are shown as below to validate that SITE has comparable accuracy as the black-box backbone on food101 data. We will also add more complex and finely trained results in the final paper.
>
> Model 				| Accuracy
> ----------------------|---------
> black-box backbone (pure ResNet-18) | 64.04%
> SITE | 63.91%
>
>
> * **SITE does not provide feature-based explanations, ...applying SITE on other domains**
>
> SITE provides interpretation to the raw pixel space, where it assigns an importance score to each pixel. This kind of interpretation is desired in CV tasks and widely studied.
>
> To apply SITE in other domains, we believe the bottleneck is finding appropriate feature extractors so that we can fit SITE to domain-specific backbone structures. Also, the type of transformations for equivariance is domain-specific. It is worth mentioning that SITE is not limited to geometric transformation (that we used in CV domain), so we believe extending SITE to other domains can be feasible. Thank you for suggesting the interesting direction! We will leave the exploration of SITE in other domains in future work.
>
> * **Computational cost of SITE compared to training a blackbox Resnet was not discussed.**
>
> We take the newest SITE model for the food101 dataset as an example since it is the most complex dataset. SITE is built to be the most complex. Here the backbone ResNet-18 has 11.2m parameters, and the autoencoder structure has 3.4m parameters. As a result, SITE has only around 1.3 times the parameters as the backbone model.
>
>
> * **Masking and measuring accuracy drop**
>
> We acknowledge that "masking and measuring accuracy drop" to evaluate interpretation is not perfect. As there is no commonly recognized "faithful interpretation" for DNNs, there is no perfect metric to measure interpretations. When conducting the experiments, we've actually considered the RemOve And Retrain (ROAR) metric, which is suggested by the reviewer. However, ROAR is arguably unfaithful, too. It can be fooled by feature correlations, as shown in [Sturmfels et al. 2020]. Besides, since ROAR requires retraining the model multiple times, it is more computationally inefficient. As a result, we finally use the "masking and measuring accuracy drop" metric for evaluation.
>
> Regarding this issue, it is worth mentioning that we have also introduced a novel metric (named self-consistent score) in Eq. (8) in the main paper to measure interpretation, which may provide a new insight on quantitatively evaluating interpretation.
>
> In order to further evaluate the interpretation, we add new experiments by using pointing game [Zhang et al. 2016] as an evaluation metric on the annotated MNIST dataset. The pointing game measures the hit ratio of interpretations, which is the higher, the better.
> The hit ratios of SITE on untransformed and transformed validation sets are **0.9993, 0.9996**, respectively. The hit ratios of all other comparing methods (included in the manuscript) are lower than 0.9. We will include more details in final paper. Similar as the experiments we conducted in the paper, all methods are trained with the same set of randomly transformed images (which includes the identity mapping, i.e., original images) and explains the same model. Thus, the significantly higher hit ratio by SITE in pointing game evaluation validated the improvement of SITE in interpretation.
>
>
> * **The relationship to previous work in Self-Interpretable Models is not clearly stated.**
>
> Just as other self-interpretable models, SITE preserves the property that it provides interpretation that reflects the true predictive mechanism of the model. SITE is different in that: 1) it has much more expressive power and can achieve comparable accuracy as a black-box model, with significantly higher accuracy than other self-interpretable models (Table 1 in main paper); 2) the interpretation from SITE is transformation equivariant, which is not yet guaranteed in other methods. Thanks for the valuable suggestion. We will include more discussion on self-interpretable models in the final paper.
>
>
> **References**
>
> [Sturmfels et al. 2020] Sturmfels, et al., "Visualizing the Impact of Feature Attribution Baselines", Distill, 2020.
>
> [Zhang et al. 2016] Zhang, Jianming, et al. "Top-down neural attention by excitation backprop." European Conference on Computer Vision. Springer International Publishing, 2016.

---

> ### Comment · Reviewer_XcMQ · 2021-08-26
> **Thank you**
>
>  I would like to thank the authors for their response. My main concern have been addressed. I increased my score from 5->6

---

> > ### Author Response · Authors · 2021-08-29
> > **Thanks for the Review**
> >
> > Thanks very much for your recognition of our work and your valuable suggestions for the revision of our paper. We really appreciate all the time and effort you put into reviewing our paper. We will add the results from the rebuttal to our final paper.

---

### Author Response · Authors · 2021-08-24
**We are happy to answer more questions if there still exist concerns for our paper.**

Dear Reviewers,

Thanks for your time and efforts in reviewing our paper. We appreciate your constructive comments. Hopefully, our response can address your concerns.

If you have further questions or confusion, we would be very happy to clarify. Thank you very much.

Best,
Authors

---

### Decision · Program_Chairs · 2021-09-27

**Decision:**

Accept (Poster)

**Comment:**

This paper considers the problem of training self-interpretable models. It first shows that existing approaches change the interpretation with input transformations even though the model is invariant to them. It then proposes a new method that fixes this problem. While the reviewers initially had concerns about the empirical evaluation, the rebuttal clarified most of the issues and all of them unanimously recommend acceptance. So, I am recommending the paper for acceptance. At the same time, there are also some concerns such as comparison with Chen et al. work that the rebuttal did not address properly. I would ask the authors to fix this and other issues raised in the reviews below in the final version.